

# Impacts of meteorology and emissions on surface ozone increases over Central Eastern China between 2003 and 2015

Lei Sun[1,2], Likun Xue[1*], Yuhang Wang[2*], Longlei Li[2], Jintai Lin[3], Ruijing Ni[3], Yingying Yan[3,4], Lulu Chen[3], Juan Li[1], Qingzhu Zhang[1], Wenxing Wang[1]

[1]Environment Research Institute, Shandong University, Ji'nan, Shandong, China

[2]School of Earth and Atmospheric Sciences, Georgia Institute of Technology, Atlanta, GA, USA

[3]Laboratory for Climate and Ocean-Atmosphere Studies, Department of Atmospheric and Oceanic Sciences, School of Physics, Peking University, Beijing, China

[4]Department of Atmospheric Sciences, School of Environmental Studies, China University of

Geosciences (Wuhan), 430074, Wuhan, China

Correspondence to:

Likun Xue (xuelikun@sdu.edu.cn) and Yuhang Wang (yuhang.wang@eas.gatech.edu)

**Abstract**

Recent studies have shown that surface ozone ($O_3$) concentrations over Central Eastern China

(CEC) have increased significantly during the past decade. We quantified the effects of changes in meteorological conditions and $O_3$ precursor emissions on surface $O_3$ levels over CEC between July 2003 and July 2015 using the GEOS-Chem model. The simulated monthly mean maximum daily 8-h average $O_3$ concentration (MDA8 $O_3$) in July increased by approximately 13.6%, from 65.5±7.9 ppbv (2003) to 74.4±8.7 ppbv (2015), comparable to the observed results. The change

in meteorology led to an increase of MDA8 $O_3$ of 5.8±3.9 ppbv over the central part of CEC, in contrast to a decrease of about -0.8±3.5 ppbv over the eastern part of the region. In comparison, the MDA8 $O_3$ over the central and eastern parts of CEC increased by 3.5±1.4 ppbv and 5.6±1.8 ppbv due to the increased emissions. The increase in regional averaged $O_3$ resulting from the emission increase (4.0±1.9 ppbv) was higher than that caused by meteorological changes

(3.1±4.9 ppbv) relative to the 2003 standard simulation, while the regions with larger $O_3$ increases showed a higher sensitivity to meteorological conditions than to emission changes. Sensitivity tests indicate that increased levels of anthropogenic non-methane volatile organic compounds (NMVOCs) dominate the $O_3$ increase over the eastern part of CEC, and anthropogenic nitrogen oxides (NOx) mainly increase $O_3$ concentrations over the central and



western parts, while decrease $O_3$ in a few urban areas in the eastern part. Process analysis showed that net photochemical production and meteorological conditions (transport in particular) are two important factors that influence $O_3$ levels over the CEC. The results of this study suggest a need to further assess the effectiveness of control strategies for $O_3$ pollution in the context of

regional meteorology, transboundary transport, and anthropogenic emission changes.

## 1. Introduction

Tropospheric ozone ($O_3$) is a major atmospheric oxidant and the primary source of hydroxyl radicals (OH), which control the atmospheric oxidizing capacity (Seinfeld and Pandis, 2016). In the troposphere, $O_3$ is produced by the photochemical oxidation of hydrocarbons, carbon

monoxide (CO) and nitrogen oxides (NOx) in the presence of sunlight, and can be transported from the stratosphere (Crutzen, 1973; Danielsen, 1968). It is an important greenhouse gas with a positive radiative forcing of 0.4 (0.2–0.6) W $m^{-2}$ (IPCC, 2013), and it has adverse effects on human health and ecosystem productivity (Monks et al., 2015).

Surface $O_3$ concentrations increased globally during the 20th century. Almost all available

monitoring data from 1950–1979 until 2000–2010 for the Northern Hemisphere indicate an increase of 1–5 ppbv per decade (Cooper et al., 2014; Monks et al., 2015), although the trends have varied regionally since the 1990s. The $O_3$ concentrations in rural and remote areas of Europe showed an increasing trend until 2000, but then tended to level off or decline (Oltmans et al., 2013; Parrish et al., 2014; Yan et al., 2018b). In the eastern US, summertime $O_3$ has

continued declining since 1990, whereas springtime $O_3$ in the western US shows large inter-annual variability (Lin et al., 2015). At some remote sites of western US, only small increases (0.00–0.43 ppbv $yr^{-1}$) have been recorded (Cooper et al., 2012). In comparison with Europe and North America, the $O_3$ concentrations in China have shown significant increasing trends since the 1990s. Ding et al. (2008) reported an increase of 3 ppbv $yr^{-1}$ in the afternoon

boundary-layer $O_3$ concentrations in summer over Beijing using aircraft data obtained by the Measurement of Ozone and Water Vapor by Airbus In-Service Aircraft (MOZAIC) program during 1995–2005. The maximum daily 8-h average $O_3$ concentration (MDA8 $O_3$) at Shangdianzi, a rural site near Beijing, showed a significant increase at a rate of about 1.1 ppbv $yr^{-1}$ from 2003 to 2015 (Ma et al., 2016). Sun et al. (2016) reported an increase of 1.7–2.1 ppbv

$yr^{-1}$ at Mt Tai during summertime from 2003 to 2015. In recent years, high $O_3$ concentrations



have been widely observed in China, especially in the Central Eastern China (CEC: 103 °E to 120 °E, 28 °N to 40 °N) during the summertime (Wang et al., 2006; 2017; Xue et al., 2014). All of these results indicate that CEC might continue to experience worsening $O_3$ air pollution. In this study, we quantify the effects of several factors on $O_3$ changes and propose some suggestions to

control surface $O_3$ in the future.

The level of $O_3$ in the troposphere is mainly determined by the abundance of its precursors, including both anthropogenic and natural emissions, and the meteorological conditions (Logan, 1985). The anthropogenic NOx emissions in China continued rising until the launch of the Twelfth Five-Year Plan (2011–2015), which enforced a series of stringent NOx emission control

measures (China State Council, 2011). However, anthropogenic emissions of non-methane volatile organic compounds (NMVOCs) continue to increase unabated (Li et al., 2017a; Zheng et al., 2018). Biomass burning also makes an important contribution to $O_3$ formation (Real et al., 2007; Yamaji et al., 2010), and biogenic emissions of isoprene and monoterpenes contribute to $O_3$ levels, which are influenced by meteorological variations (Fu and Liao, 2012). Meteorological

parameters, such as wind, temperature and humidity, can influence $O_3$ concentrations via mechanisms related to transport, chemical production and loss, and deposition (Monks, 2000; Zhao et al., 2010). Studies in the past two decades have shown that $O_3$ and its precursors can be transported across regions and even hemispheres, as it has a lifetime of several days to weeks in the troposphere (Jacob et al., 1999; Lin et al., 2008; Verstraeten et al., 2015). For example, Ni et

al. (2018) showed significant foreign contributions to springtime $O_3$ over China. In addition, the stratosphere–troposphere exchange (STE) is another important process affecting the tropospheric $O_3$ burden, especially in the mid-latitudes of the Northern Hemisphere during springtime (Hess and Zbinden, 2013). However, currently there is still large variation in quantifying the contribution of each factor to the $O_3$ trends among different models and study regions (Zhang et

al., 2014a).

Previous studies have revealed the important effects of changing emission levels and varying climate conditions on tropospheric $O_3$ in different regions. Lou et al. (2015) found that the effect of variations in meteorological conditions on the inter-annual variability of surface $O_3$ was larger than that of variations in anthropogenic emissions in Eastern China from 2004–2012. Using the

GEOS-Chem model, Yan et al. (2018a) found that inter-annual climate variability is the main driver of daytime $O_3$ variability in the US, although the reduction of anthropogenic emissions of



NOx increased the night-time $O_3$ concentrations due to reduced $O_3$ titration. The effects of East Asian summer monsoon on surface $O_3$ have been analyzed by observational and modeling studies (He et al., 2008; Wang et al., 2011; Zhao et al., 2010). Given the scarcity of previous research, it is necessary to further quantify the contributions of emissions and meteorological

conditions to surface $O_3$ levels to deepen our understanding of the factors influencing $O_3$ changes in China.

This study integrates the global GEOS-Chem model and its Asia nested model to investigate the spatial distributions of surface $O_3$ over CEC in July 2003 and July 2015. Meteorological conditions and $O_3$ precursor emissions are identified as the dominant factors influencing $O_3$

levels, and their contributions are quantified. We identify the key factors that affect $O_3$ changes and make a policy recommendation for $O_3$ control over CEC in the future. Section 2 briefly introduces the GEOS-Chem model and simulation scenarios. Comparisons of the simulated and observed $O_3$ concentrations are made in Section 3. We quantify the individual effects of meteorological conditions and emissions on $O_3$ changes in Section 4 and Section 5, respectively.

In Section 6, we discuss important processes influencing $O_3$ changes. Section 7 concludes the paper.

## 2. Model and simulations

### 2.1 Model description

A nested model coupled with the global chemical transport model GEOS-Chem v11-01

(http://acmg.seas.harvard.edu/geos/doc/man/), is used to simulate the surface $O_3$ concentrations and distributions over CEC in July of 2003 and 2015. The meteorological field is taken from MERRA-2 as assimilated by the Goddard Earth Observing System (GEOS) at NASA's Global Modelling and Assimilation Office. The global model and its nested model, covering China and South East Asia (60 °E to 150 °E, 11 °S to 55 °N), are configured to have horizontal spatial

resolutions of $2° \times 2.5°$ and $0.5° \times 0.625°$, respectively, by latitude and longitude, and 47-layer reduced grids in the vertical direction with 10 layers (each ~130 m in thickness) below 850 hPa. The models are run with the full standard NOx-Ox-hydrocarbon-aerosol tropospheric chemistry (Mao et al., 2013) for January to July of 2003 and 2015, including the spin-up time, but only the results for July are analysed. We use the Linoz stratospheric ozone chemistry mechanism for

stratospheric $O_3$ production (McLinden et al., 2000), and the non-local planetary boundary layer



(PBL) mixing scheme for vertical mixing of air tracers in the PBL (Holtslag and Boville, 1993; Lin and McElroy, 2010).

Global anthropogenic emissions of NOx and CO for 2003 and 2008 are taken from EDGAR v4.2 (Emission Database for Global Atmospheric Research, http://edgar.jrc.ec.europa.eu/overview.php?v=42). NMVOC emissions are taken from the RETRO (REanalysis of TROpospheric chemical composition) inventory for 2000, but the emissions of $C_2H_6$ and $C_3H_8$ follow Xiao et al. (2008). For Europe, the United States, Asia, China, Canada and Mexico, the anthropogenic emissions are taken from EMEP (from 2003 to 2012; Auvray et al., 2005), NEI2011 (base year: 2011, annual scale factors: 2006–2013; ftp://aftp.fsl.noaa.gov/divisions/taq/), MIX (from 2008 to 2010; Li et al., 2017b), MEIC (2008 and 2014, http://meicmodel.org), CAC (NOx and CO: from 2003 to 2008 (scaled to 2010); http://www.ec.gc.ca/pdb/cac/cac_home_e.cfm), and BRAVO (1999; Kuhns et al., 2003), respectively. Over China, the CO, NOx and NMVOC emissions from MEIC for 2008 are scaled to 2003 based on the inter-annual variability of Regional Emission in Asia (REAS-v2; Kurokawa et al., 2013), but the anthropogenic emissions for 2014 are taken directly without being scaled to 2015. According to Zheng et al. (2018), the anthropogenic NOx and NMVOC emissions in China decreased by about 6% and 2% from 2014 to 2015, respectively, so here we may slightly overestimate the NOx and NMVOC emissions. Daily biomass burning emissions are taken from the Global Fire Emission Database v4 (GFED4) (Randerson et al., 2012). Biogenic emissions in the GEOS-Chem model are calculated online from the MEGAN v2.1 scheme (Guenther et al., 2012). Natural NOx emissions from lightning are parameterised following Price and Rind (1992), and are further constrained by the LIS/OTD satellite data (Murray et al., 2012). We obtain the vertical profile of the lightning NOx based on Ott et al. (2010) and calculate the soil NOx emissions online following Hudman et al. (2012).

**2.2 Model simulations**

Table 1 summarises the six model scenarios we set to identify the contributions from the changes in meteorological conditions and emissions between 2003 and 2015. We refer to the scenario using the emissions described in the previous section as the standard simulation, and define the standard simulations for 2003 and 2015 as 03E03M and 15E15M (2003 emissions + 2003 meteorology and 2015 emissions + 2015 meteorology). In this case, the difference between



O$_3$ concentrations for 03E03M and 15E15M (denoted as 15E15M-03E03M) is due to the combined effect of changes in emissions and meteorology between 2003 and 2015. Similarly, scenarios with 2003 emissions + 2015 meteorology and 2015 emissions + 2003 meteorology are defined as 03E15M and 15E03M, respectively. The contribution of the change in meteorological conditions can thus be calculated by the difference between the simulated O$_3$ concentrations in the 03E15M and 03E03M scenarios (03E15M-03E03M). Similarly, the contribution of emission changes can be calculated by 15E03M-03E03M (or 15E15M-03E15M). The contribution of the meteorological change based on the 2015 standard simulation is given by 15E15M-15E03M. Since the amount of O$_3$ formed responds nonlinearly to the NOx and NMVOC emissions, the sum of (03E15M-03E03M) and (15E03M-03E03M) does not equal to (15E15M-03E03M). However, we can still compare these two scenarios to quantify the effects of meteorology and emission changes.

We then investigate the effect of anthropogenic emissions (NOx and NMVOCs) on surface O$_3$ concentrations based on the 2015 simulations. We replace the anthropogenic NOx or NMVOC emissions in the 2015 standard simulation with corresponding emissions for 2003 and keep the meteorology field, biomass burning, and natural emissions (NOx from soil and lightning, biogenic VOCs (BVOCs), etc.) unchanged (03N15M and 03V15M, respectively). The contribution of anthropogenic NOx (NMVOCs) emission changes can be calculated by the difference between the 2015 standard simulation and 03N15M (03V15M), defined as 15E15M-03N15M (15E15M-03V15M).

## 3. Simulated and observed O$_3$ concentrations

### 3.1 Model evaluation

In this section, we evaluate the model's performance by comparing the simulated surface O$_3$ concentrations with observations from rural/regional background sites and the network of the Chinese National Environmental Monitoring Center (http://datacenter.mep.gov.cn/).

For 2003, only a few non-urban sites over CEC have surface O$_3$ measurements available. We selected six rural/regional background sites for the model evaluation: Mt Tai (36.25 °N, 117.10 °E; 1534 m a.s.l.), Mt Hua (34.49 °N, 110.09 °E; 2064 m a.s.l.), Mt Huang (30.13 °N, 118.15 °E; 1840 m a.s.l.), Shangdianzi (SDZ: 40.65 °N, 117.12 °E; 293 m a.s.l.), Lin'an (30.30 °N, 119.73 °E; 139 m a.s.l.), and Hok Tsui (22.22 °N, 114.25 °E; 60 m a.s.l.) (see Fig. S1 for the locations of these





sites). The monthly mean $O_3$ concentrations at these six sites were taken from the literature (Li et al., 2007; Meng et al., 2009; Wang et al., 2009; Fan et al., 2013; Sun et al., 2016). We compare the simulated surface $O_3$ concentrations with the 2003 observations for Mt Tai and Hok Tsui but with the 2004 observations for the other four sites.

Figure 1(a) compares the observed and simulated monthly mean $O_3$ concentrations at the six sites. The simulated $O_3$ concentrations match the observations at Mt Tai, SDZ, and Mt Hua well, with only minor positive biases (1–4 ppbv). In contrast, the model overestimates the $O_3$ concentrations at Mt Huang, Lin'an, and Hok Tsui by approximately 10 ppbv. These sites in the south sector are often rainy or cloudy during summer, so the overestimation of $O_3$ is likely to be

due to the model's underestimation of precipitation and cloud cover (Ni et al., 2018). The overestimation at the Hok Tsui coastal site of Hong Kong also reflects that the model resolution is insufficient to capture the local terrains and transport pathway (Ni et al., 2018). Similar results were obtained from the comparison between observed and simulated monthly mean $O_3$ concentrations at the six sites in July 2004 (see Figure S2).

For 2015, the simulated $O_3$ concentrations are compared with observations by the network of the Chinese National Environmental Monitoring Center over East China (Figure 1(b)). To avoid contamination by local pollutants, we select 115 non-urban sites located in 115 cities of East China. For cities in which no non-urban sites are available, we choose sites that are least affected by local pollution (i.e., sites relatively far away from roads, factories, power plants, etc.). For

MDA8 $O_3$, the model results are highly correlated with the observations at most sites ($R^2 = 0.79$). The model only overestimates the monthly MDA8 $O_3$ by approximately 2.7±5.9 ppbv over CEC.

The model also captures the spatial distribution of MDA8 $O_3$ very well. It ranges from 40–60 ppbv in the south to 80–100 ppbv in the north of CEC (Figure 2(b)), similar to patterns reported by Lin et al. (2009) and Lou et al. (2015). Hourly $O_3$ concentrations from the model and

observations at Mt Tai in 2003 and nine representative sites in 2015 are compared in Figures S3 and S4, respectively. The model reproduces the diurnal variations in $O_3$ with a normalised mean bias of 4% at Mt Tai. For the nine sites, the model captures most day-to-day variability (Figure S4). However, it produces larger biases during the night, mostly due to the titration of NO and a lower inversion layer (Yan et al., 2018a). The overestimation of $O_3$ concentrations in the

afternoon is likely to be due to the overestimated precursor emissions in the model.





The observed MDA8 $O_3$ at SDZ station increased by about 10.8 ppbv from July 2004 to July 2014, comparable to the simulated result, which showed an increase of about 9.5 ppbv from July 2003 to July 2015. In addition, the increasing trend of observed $O_3$ at Mt Tai was 2.2 ppbv $yr^{-1}$ from 2003 to 2015, higher than the simulated increase of about 1.3 ppbv $yr^{-1}$. Nonetheless, the

model captures the significant increase in surface $O_3$ levels over CEC between July 2003 and July 2015.

## 3.2 Spatial distribution and diurnal variation simulated in different model scenarios

Figure 2 shows the simulated spatial distribution of monthly mean surface MDA8 $O_3$ over eastern China (100 °E to 125 °E, 20 °N to 50 °N) for July 2003 and July 2015. The model simulates

relatively high $O_3$ concentrations over the North China Plain and Sichuan Basin, where anthropogenic emissions of $O_3$ precursors are high. In July 2003, only a small area in CEC had an MDA8 $O_3$ concentration exceeding the Level II National Ambient Air Quality Standard (75 ppbv) (Figure 2(a)), but in July 2015 it had expanded to nearly half of this region. The regional mean MDA8 $O_3$ increased from 65.5±7.9 ppbv in July 2003 to 74.4±8.7 ppbv in July 2015

(Table 2). This increasing rate (0.74 ppbv $yr^{-1}$) is slightly smaller than the findings of most studies (positive trends: 1–3 ppbv $yr^{-1}$) in this region over the past decade (Ding et al., 2008; Ma et al., 2016; Sun et al., 2016; Xu et al., 2008; Zhang et al., 2014b). Both daily mean and MDA8 $O_3$ concentrations were significantly higher in July 2015 than in July 2013 over most areas of CEC (Figure 3). As the concentrations of MDA8 $O_3$ over southwestern China did not exceed the

Level II National Ambient Air Quality Standard in July 2015, we do not focus our analysis on this area in the following sections.

The diurnal variation of $O_3$ over CEC illustrated in Figure 4 shows that $O_3$ increases by 4.9–6.7 ppbv before dawn (02:00–07:00) and by 8.5–9.0 ppbv in the afternoon (13:00–18:00). The much more significant increase of $O_3$ in the afternoon in July 2015 is likely to be due to the stronger

photochemical production, which is affected by both meteorological conditions and $O_3$ precursor emissions. The slight increase in night-time $O_3$ reflects the residual effect of the daytime increase, despite strong night-time titration by NO. This result is very different from the trends over the US, where summertime daytime $O_3$ increased over the past decades is contrast to the night-time decrease in all seasons (Yan et al., 2018a). Therefore, we focus on the MDA8 $O_3$ changes over

CEC between July 2003 and July 2015.



## 4. Impacts of meteorology on surface $O_3$

We performed sensitivity tests to investigate the effects of meteorology and emissions on the MDA8 $O_3$ over CEC. The contributions of meteorological change to the change in MDA8 $O_3$ are defined by the 03E15M-03E03M and 15E15M-15E03M simulations. Here we discuss only

03E15M-03E03M in detail, as the results of 15E15M-15E03M are similar.

The regional averaged MDA8 $O_3$ simulated by 03E15M is 68.7±7.1 ppbv, comparable with that simulated by 15E03M (69.6±8.9 ppbv), indicating the comparable contributions made by the changes in meteorology and in emissions. Figure 5 shows the spatial distribution of MDA8 $O_3$ changes between different simulation scenarios. The domain-averaged MDA8 $O_3$ is

approximately 5.8±3.9 ppbv (5%–95% interval: -0.1–12.4 ppbv) higher in scenario 03E15M than in 03E03M (Figure 5(a)) over the central part of CEC (106 °E to 115 °E, 28 °N to 40 °N). Over the eastern coastal areas (115 °E to 120 °E, 28 °N to 40 °N), however, the MDA8 $O_3$ in the former scenario is less than in the latter by approximately -0.8±3.5 ppbv (5%–95% interval: -6.8–3.8 ppbv), indicating great spatial variation in the influence of meteorological changes.

Atmospheric circulation patterns complicate the prediction of $O_3$ concentrations in a specific region (He et al., 2012). The geopotential height map in Figure S5 shows a high-pressure system over CEC at 850 hPa in July 2015. It is well known that high $O_3$ pollution events preferentially occur under high-pressure conditions (Wild et al., 2004; Zhao et al., 2009; Xu et al., 2011). This is because the relatively high geopotential height induces a stable weather condition. Neither

horizontal nor vertical transport is strong, which favours the accumulation of atmospheric pollutants such as surface $O_3$. We found that in July 2015, the wind speeds over southern and eastern boundaries of CEC were much slower than that in July 2003 (Figure S6), leading to much lower $O_3$ flux across these two boundaries. The low $O_3$ over southern CEC in July 2003 was mainly due to the strong south-westerly wind, decreasing $O_3$ levels in this area. However, a

large amount of $O_3$ and its precursors from the central part of CEC were transported to the eastern coastal area, which increased $O_3$ concentrations there (refer to Table 4: about 1343 Gg mon$^{-1}$ $O_3$ transported out across the east boundary). Conversely, in July 2015, only a small amount of $O_3$ (refer to Table 4: -61 Gg mon$^{-1}$) and its precursors were transported away from the ocean by the weak south-easterly winds, which only decreased the $O_3$ levels in the coastal area.

However, in the central part of CEC, the wind was weak, leading to accumulating $O_3$ pollution in



this area. As a result, the $O_3$ concentrations increased in the central part of CEC and decreased in the eastern coastal area in July 2015 compared to July 2003.

In addition to the wind, air temperature and relative humidity are two other important meteorological parameters that can affect atmospheric $O_3$ concentrations. High temperatures tend
to accelerate the rate of ozone-related photochemical reactions, promoting $O_3$ production (Ramsey et al., 2014). Cloud indirectly affects $O_3$ pollution by blocking solar radiation, thus affecting the emission of BVOCs and the photochemical production of $O_3$ (Lin et al., 2009). Neither air temperature nor relative humidity plays an important role in explaining the difference in surface $O_3$ between 2003 and 2015; they are at almost the same levels (Figure S7). The
average net production of $O_3$ over CEC simulated by 03E03M (11.7 ppbv day$^{-1}$) is very close to that simulated by 03E15M (11.9 ppbv day$^{-1}$) (Table 4), suggesting that meteorological factors in 2003 and 2015 did not greatly change $O_3$ photochemical reactions.

Table 2 shows the monthly mean MDA8 $O_3$ over CEC. We summarise the regional mean $O_3$ over CEC and regions with MDA8 $O_3$ >75 ppbv in July 2015. To avoid the influence of uneven
spatial distributions of $O_3$ concentration changes, we performed a gradient analysis. The differences in MDA8 $O_3$ were analysed in four ways: regional mean, $\Delta$MDA8 $O_3$ $\geqslant$0 ppbv (the region over which the difference of MDA8 $O_3$ between the 2003 standard and 2015 standard simulation (15E15M-03E03M) is positive), $\Delta$MDA8 $O_3$ $\geqslant$5 ppbv and $\Delta$MDA8 $O_3$ $\geqslant$10 ppbv. For the regional mean over CEC, the increase in $O_3$ concentration driven by meteorology is
approximately 3.1±4.9 ppbv, from 65.5±7.9 ppbv (03E03M) to 68.7±7.1 ppbv (03E15M). Where $\Delta$MDA8 $O_3$ $\geqslant$10 ppbv, mostly over the central part of CEC, the $O_3$ concentration increases by 6.7±3.4 ppbv from 64.3±9.7 ppbv to 71.0±7.4 ppbv due to the meteorological change. Thus, the meteorological conditions have a greater impact on the $O_3$ change when the difference between 2003 and 2015 is higher than 10 ppbv. Similar results are also found in regions with MDA8
$O_3$ >75 ppbv, where the increase in the $O_3$ concentration is approximately 3.6±3.2 ppbv and 5.1±2.5 ppbv for the regional mean and for the $\Delta$MDA8 $O_3$ $\geqslant$10 case, respectively. This indicates that surface $O_3$ levels are more sensitive to meteorological conditions in regions with larger $O_3$ increase.

## 5. Impact of emission changes on surface $O_3$





As described above, the impact of emission changes on MDA8 $O_3$ concentrations between 2003 and 2015 can be estimated by 15E03M-03E03M or 15E15M-03E15M. Here we discuss only 15E03M-03E03M in detail. Similar results were found from 15E15M-03E15M.

Figure 5(b) shows the contributions of emission changes to surface $O_3$ levels. The emission
change leads to an increase in MDA8 $O_3$ over most areas of CEC, and it has a much smaller spatial variability than the meteorological change does (Figure 5(a)). Compared to the influence of the meteorological change (03E15M-03E03M: 3.1±4.9 ppbv), the increase in emissions leads to a higher regional mean $O_3$ increase (15E03M-03E03M: 4.0±1.9 ppbv) over CEC (Table 2). In contrast, for the case of $\Delta$MDA8 $O_3 \geqslant 10$ ppbv, the influence of emission change on $O_3$
(15E03M-03E03M: 4.5±2.1 ppbv) is smaller than that of the meteorological field change (03E15M-03E03M: 6.7±3.4 ppbv). The increases of MDA8 $O_3$ due to emission change are about 3.5±1.4 ppbv (5%–95% interval: 1.6–6.0 ppbv) and 5.6±1.8 ppbv (5%–95% interval: 2.2–8.4 ppbv) over the central and eastern parts of CEC, which are different with the spatial pattern caused by meteorological condition change. It is worth noting that in the polluted regions where
MDA8 $O_3 >75$ ppbv, even if the $\Delta$MDA8 $O_3$ is greater than 10 ppbv, the $O_3$ increase caused by emission change is still higher than that caused by meteorological conditions, indicating the dominant effect of emissions on $O_3$ pollution in the highly polluted regions.

We summarise the emissions of NOx, CO and NMVOCs over CEC for July 2003 and July 2015 in Table 3. The anthropogenic NOx emissions increased from 397 Gg mon$^{-1}$ in July 2003 to
683 Gg mon$^{-1}$ in July 2015. The anthropogenic NMVOCs also increased significantly, with the NMVOC emissions increasing from 190 Gg C mon$^{-1}$ in July 2003 to 365 Gg C mon$^{-1}$ in July 2015. Anthropogenic CO emissions increased from 4619 Gg mon$^{-1}$ in July 2003 to 6011 Gg mon$^{-1}$ in July 2015. The natural BVOCs, which are greatly affected by meteorological conditions, remained unchanged between 2003 and 2015. Biomass burning often occurs sequentially from
south to north in CEC in the spring harvest season and lasts from late May to mid-June (Chen et al., 2017). In July, the biomass burning emissions generally decrease to approximately 1% of the anthropogenic emissions (not shown). Therefore, the effect of the emission change on $O_3$ is primarily due to anthropogenic emissions of NOx and NMVOCs.

To separate the effect of anthropogenic emissions from the effect of natural emission on $O_3$
variability, we conducted two further simulations, 03N15M and 03V15M (see Section 2.2).





Figure 6 shows the spatial distribution of the MDA8 $O_3$ differences between the 2015 standard simulation and these two simulations. Anthropogenic NMVOCs (Figure 6 (a)) have a great impact on MDA8 $O_3$ over the eastern part of CEC, increasing its concentration by approximately 2.5±0.8 ppbv (5%–95% interval: 1.1–3.7 ppbv). The change in $O_3$ concentrations due to

anthropogenic NMVOCs varies from -0.5 ppbv to 5.1 ppbv over different sub-regions of CEC, with a regional mean of 1.4±1.1 ppbv. The effect of anthropogenic NOx (Figure 6 (b)), in comparison, is more complicated. From 2003 to 2015, MDA8 $O_3$ declined in some cities such as Tianjin, Ji'nan, Taiyuan, and Nanjing in the eastern part of CEC, but increased in the central and western parts (domain mean: 2.8±0.9 ppbv, 5%–95% interval: 1.4–4.1 ppbv). The change in

MDA8 $O_3$ due to anthropogenic NOx varies from -3.1 ppbv to 6.7 ppbv, with a regional mean of 2.5±1.1 ppbv over CEC (5%–95% interval: -0.2–3.3 ppbv). The reduction of $O_3$ in the urban area is likely to be due to the abundant NOx from industrial and traffic sources. Beijing shows a slight decrease in NOx and NMVOC emissions, leading to a slight change in $O_3$ levels. In most rural areas of CEC, $O_3$ formation tends to be limited by the concentrations of NOx (the so-called

NOx-limited regime). Thus, $O_3$ is increased significantly as we increase the anthropogenic emissions of NOx. A VOC-limited regime in a few urban areas and a NOx-limited/transition regime in regional rural areas of CEC have been reported in some observational and model simulation studies (Wang et al., 2017 and references therein). The change in BVOC emissions only leads to a small change in MDA8 $O_3$ over CEC, resulting in an increase in the $O_3$ level of

only 0.3 ppbv (not shown), mostly due to the change in meteorological conditions. Therefore, if the meteorological conditions are fixed as the 2015 conditions, the increase in anthropogenic NMVOCs is the most important factor responsible for the in $O_3$ increase over the eastern part of CEC, whereas NOx emissions tend to increase $O_3$ concentrations over central and western parts but decrease it in a few urban areas over eastern parts of CEC.

**6. Process analyses**

Ozone concentrations are determined by chemical and dynamic processes including transport, chemical production and loss, and deposition. In this section, we discuss the effect of these processes on the surface $O_3$ over CEC.

Table 4 documents the horizontal and vertical mass fluxes of $O_3$ over CEC at four boundaries

(north, east, south and west). The flux at each boundary was calculated from surface to 850 hPa.



In July 2003, the air flows into CEC through the south boundary, and then out across the other three boundaries. In contrast, the air masses flow into this area across the east boundary in July 2015, and then out across the left three boundaries. The larger $O_3$ flux from each boundary in July 2003 is due to stronger winds. Compared to the 03E03M simulation (-897 Gg mon$^{-1}$; minus value means export of $O_3$ from this region), 03E15M shows a much lower $O_3$ flux (-401 Gg mon$^{-1}$), indicating that weather conditions in 2015 play a more important role in pollutant accumulation, which is consistent with our analysis in Section 4. The larger $O_3$ flux in 15E03M (-1232 Gg mon$^{-1}$) in comparison to the 03E03M simulation, however, is mostly due to the increased precursor emissions in 2015.

Table 4 also shows the chemical production and loss of $O_3$ over CEC from surface to 850 hPa. The net photochemical production of $O_3$ in July 2015 (2158 Gg mon$^{-1}$ or 15.5 ppbv day$^{-1}$) is higher than that in July 2003 (1629 Gg mon$^{-1}$ or 11.7 ppbv day$^{-1}$). By comparing the 03E03M simulation with 03E15M simulation, we find that the weather conditions in 2015 do not promote excessive $O_3$ production (03E15M: 1657 Gg mon$^{-1}$ or 11.9 ppbv day$^{-1}$), almost the same level as 03E03M simulation. In comparison, due to more $O_3$ precursor emissions in 2015, the $O_3$ production by 15E03M (2166 Gg mon$^{-1}$ or 15.6 ppbv day$^{-1}$) is much higher than the 03E03M simulation. The net photochemical $O_3$ production in this study is similar to the result of Li et al. (2007), who reported a net production of 10–32 ppbv day$^{-1}$ at three mountain sites over CEC in 2004. Deposition (mainly dry deposition) is another factor that affects $O_3$ concentrations. The 03E15M simulation shows an increase in $O_3$ dry deposition by only 10 Gg mon$^{-1}$, compared to the 03E03M simulation (156 Gg mon$^{-1}$). So dry deposition is less affected by changes in weather conditions.

As shown in Table 4, the $O_3$ budget analysis indicates CEC is a strong photochemical source region in both 2003 and 2015. The photochemically produced $O_3$ is exported by transport and to a lesser extent removed by dry deposition. In July 2003, about half of photochemically formed $O_3$ in the CEC region was removed by transport in July 2015. Comparing the results of the 2003 and 2015 standard simulations (15E15M-03E03M), we find that the absolute value of $O_3$ transport flux increased by 395 Gg mon$^{-1}$ (2015-2003), net $O_3$ production increased by 529 Gg mon$^{-1}$ and $O_3$ dry deposition only increased by 24 Gg mon$^{-1}$. As a result, the increase in $O_3$ concentrations from July 2003 to July 2015 is mainly due to transboundary horizontal transport, vertical transport and photochemical reactions.





## 7. Conclusions

In this study, we used the global GEOS-Chem model and its Asia nested model to simulate surface $O_3$ over Central Eastern China between July 2003 and July 2015. We found that the regional averaged concentration of MDA8 $O_3$ increased from 65.5±7.9 ppbv in 2003 to 74.4±8.7 ppbv in 2015. The increase in the regional average MDA8 $O_3$ due to emission changes (4.0±1.9 ppbv) is higher than that caused by meteorological changes (3.1±4.9 ppbv) compared with the 2003 standard simulation. The effects of meteorological changes have a larger spatial variability than those of emission changes. The increase in anthropogenic NMVOC emissions increased $O_3$ concentrations over the eastern part of CEC, whereas the increased anthropogenic NOx emissions dominated the increase in $O_3$ over the central and western parts of CEC but decreased $O_3$ levels in a few urban areas over eastern CEC. The $O_3$ formation over most areas is in NOx-limited or transition regime, whereas a few urban areas tend to be in VOC-limited regime. The increase in surface $O_3$ concentrations is mainly via photochemical production and transport processes. The transport pattern in July 2015 tends to enhance $O_3$ levels over the central part of CEC, while the meteorological condition in 2015 does not promote the $O_3$ production. The increased net $O_3$ photochemical production is mostly due to increased precursor emissions.

Our results have implications for the formulation of effective control strategies for $O_3$ air pollution in CEC. Although the simulated average effect of emission changes is larger than the effect of meteorological changes, the regions with larger $O_3$ increases (e.g., ΔMDA8 $O_3$ ≥10 ppbv) show a higher sensitivity to meteorology than to emission changes. The results imply that assessment of the effectiveness of regional and urban $O_3$ control strategies needs to be placed in the context of meteorology. The $O_3$ transport flux analysis further suggests that large-scale regional transport is an important contributor to the surface $O_3$ increases from 2003 to 2015. Transboundary transport issues in local $O_3$ control strategies should go beyond transport from neighbouring areas and account for long-distance transport, particularly in the context of globalizing air pollution.

## Acknowledgements

This research was supported by the National Key Research and Development Program of China (2016YFC0200500), the National Natural Science Foundation of China (41675118, 91544213, 41775115), the Qilu Youth Talent Program of Shandong University, the Jiangsu



Collaborative Innovation Center for Climate Change, and the Taishan Scholars (ts201712003). The model simulations were done at the Supercomputing Center of Shandong University in Weihai. We thank the Chinese National Environmental Monitoring Center for providing the observation data. Lei Sun acknowledges the support of the China Scholarship Council.

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



**Table 1.** Model simulation scenarios in this study.

| Name | Description |
| --- | --- |
| 1. 2003 standard (03E03M) | The standard simulation of $O_3$ concentrations over China based on 2003 emissions and 2003 meteorology |
| 2. 2015 standard (15E15M) | The standard simulation of $O_3$ concentrations over China based on 2015 emissions and 2015 meteorology |
| 3. 03E15M | Same as 2 but with 2003 emissions |
| 4. 15E03M | Same as 2 but with 2003 meteorology |
| 5. 03N15M | Same as 2 but with 2003 anthropogenic NOx emissions in China |
| 6. 03V15M | Same as 2 but with 2003 anthropogenic NMVOC emissions in China |

5   **Table 2.** Monthly mean (standard deviation) MDA8 $O_3$ over CEC based on four model simulations. $\Delta$MDA8 $O_3$ represents the difference in $O_3$ concentrations between the 2015 standard simulation and 2003 standard simulation: $\Delta$MDA8 $O_3$ = MDA8 $O_3$ (2015) - MDA8 $O_3$ (2003). MDA8 $O_3$ > 75 ppbv indicates the region of MDA8 $O_3$ exceeding the Level II National Ambient Air Quality Standard (75 ppbv) in July 2015.

| Region | Description | 03E03M | 03E15M | 15E03M | 15E15M |
| --- | --- | --- | --- | --- | --- |
| CEC | regional mean | 65.5 (7.9) | 68.7 (7.1) | 69.6 (8.9) | 74.4 (8.7) |
| | $\Delta$MDA8 $O_3$ $\geqslant$0 | 65.6 (8.2) | 69.4 (6.9) | 69.8 (9.1) | 75.6 (8.2) |
| | $\Delta$MDA8 $O_3$ $\geqslant$5.0 | 65.6 (8.7) | 70.6 (6.7) | 70.0 (9.7) | 77.4 (7.7) |
| | $\Delta$MDA8 $O_3$ $\geqslant$10 | 64.3 (9.7) | 71.0 (7.4) | 68.8 (10.8) | 78.0 (8.4) |
| Region with MDA8 $O_3$ > 75 ppbv | regional mean | 71.0 (4.5) | 74.7 (4.2) | 76.0 (5.2) | 82.2 (4.7) |
| | $\Delta$MDA8 $O_3$ $\geqslant$0 | 71.0 (4.5) | 74.7 (4.2) | 76.0 (5.2) | 82.2 (4.7) |
| | $\Delta$MDA8 $O_3$ $\geqslant$5.0 | 70.9 (4.5) | 74.7 (4.2) | 75.9 (5.3) | 82.3 (4.7) |
| | $\Delta$MDA8 $O_3$ $\geqslant$10 | 70.5 (4.8) | 75.5 (4.5) | 75.7 (5.6) | 83.4 (4.8) |



**Table 3.** Emissions of NOx, CO and NMVOCs in CEC for July 2003 and July 2015, including anthropogenic emissions and biogenic emissions. Units: NO, CO and CH$_2$O: Gg mon$^{-1}$; others: Gg C mon$^{-1}$.

| Species | 2003 | 2015 | Species | 2003 | 2015 |
|---|---|---|---|---|---|
| **Anthropogenic emissions** | | | | | |
| NO | 397 | 683 | Acetaldehyde | 2.7 | 3.0 |
| CO | 4619 | 6011 | PRPE[b] | 39 | 70 |
| ALK4[a] | 81 | 184 | C$_3$H$_8$ | 35 | 56 |
| Acetone | 4.2 | 9.9 | CH$_2$O | 6.4 | 7.4 |
| Methyl Ethyl Ketone | 1.2 | 3.6 | C$_2$H$_6$ | 21 | 32 |
| **Biogenic emissions** | | | | | |
| Isoprene | 276 | 275 | b-Pinene | 18.4 | 17.4 |
| Acetone | 23.0 | 22.0 | 3-Carene | 15.3 | 14.3 |
| PRPE | 21.0 | 21.0 | Ocimene | 7.3 | 7.1 |
| a-Pinene | 25.9 | 23.8 | Acetaldehyde | 10.0 | 9.0 |
| Total monoterpenes | 90 | 85 | Other monoterpenes | 11.0 | 11.0 |

5  [a]:ALK4: Alkanes and other non-aromatic compounds that react only with OH, and have k$_{OH}$ between $5 \times 10^3$ and $1 \times 10^4$ ppm$^{-1}$ min$^{-1}$.
[b]:PRPE: OLE1+OLE2, OLE1: Alkenes (other than ethene) with k$_{OH}$ <$7 \times 10^4$ ppm$^{-1}$ min$^{-1}$; OLE2: Alkenes with k$_{OH}$ >$7 \times 10^4$ ppm$^{-1}$ min$^{-1}$.

**Table 4.** Horizontal and vertical flux (Gg mon$^{-1}$), photochemical production and loss (Gg mon$^{-1}$ (ppbv day$^{-1}$)), and dry deposition (Gg mon$^{-1}$) of O$_3$ over CEC from surface to 850 hPa based on four types of simulations. For horizontal flux, positive values indicate eastward or northward transport. For vertical fluxes, positive values indicate upward transport. Net photochemical O$_3$
15  production is the difference between production and loss of O$_3$.

| Processes | | Boundary | 03E03M | 03E15M | 15E03M | 15E15M |
|---|---|---|---|---|---|---|
| Transport | Horizontal | 105 °E | -176 | -145 | -190 | -149 |
| | | 122 °E | 1343 | -129 | 1450 | -61 |
| | | 28 °N | 1914 | -100 | 1906 | -178 |
| | | 41 °N | 440 | 327 | 488 | 351 |
| | Vertical | 850 hPa | 852 | -43 | 877 | -116 |
| | Total | | -897 | -401 | -1100 | -502 |
| Photochemical | Production | | 2850 (20.5) | 2890 (20.7) | 3511 (25.2) | 3532 (25.4) |
| | Loss | | 1221 (8.8) | 1232 (8.9) | 1344 (9.7) | 1373 (9.9) |
| | Net | | 1629 (11.7) | 1657 (11.9) | 2166 (15.6) | 2158 (15.5) |
| Dry deposition | | | 156 | 166 | 162 | 180 |





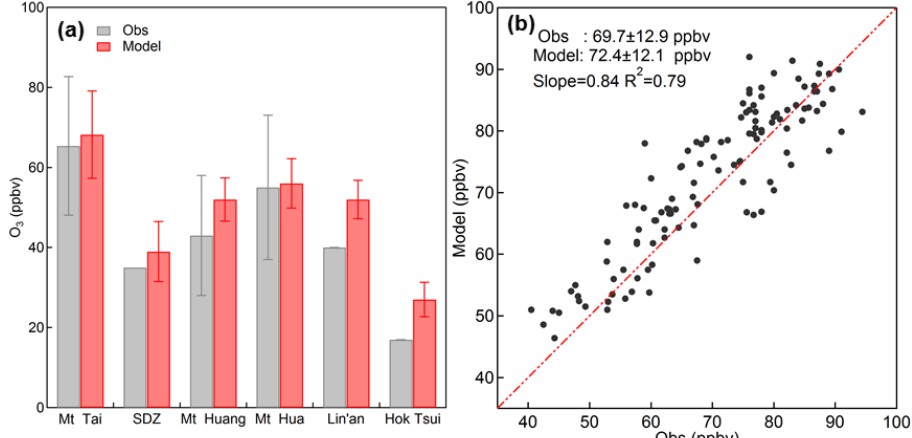

**Figure 1.** (a) Comparison of observed and simulated monthly mean concentrations of surface O$_3$ in July 2003. (Mt Tai: July 2003; SDZ: Shangdianzi station: July 2004; Mt Huang: July 2004; Mt Hua: July 2004; Lin'an: July 2004 and Hok Tsui: July 2003). (b) Correlation between observed and modelled monthly mean MDA8 O$_3$ in July 2015 at 115 stations in Eastern China.



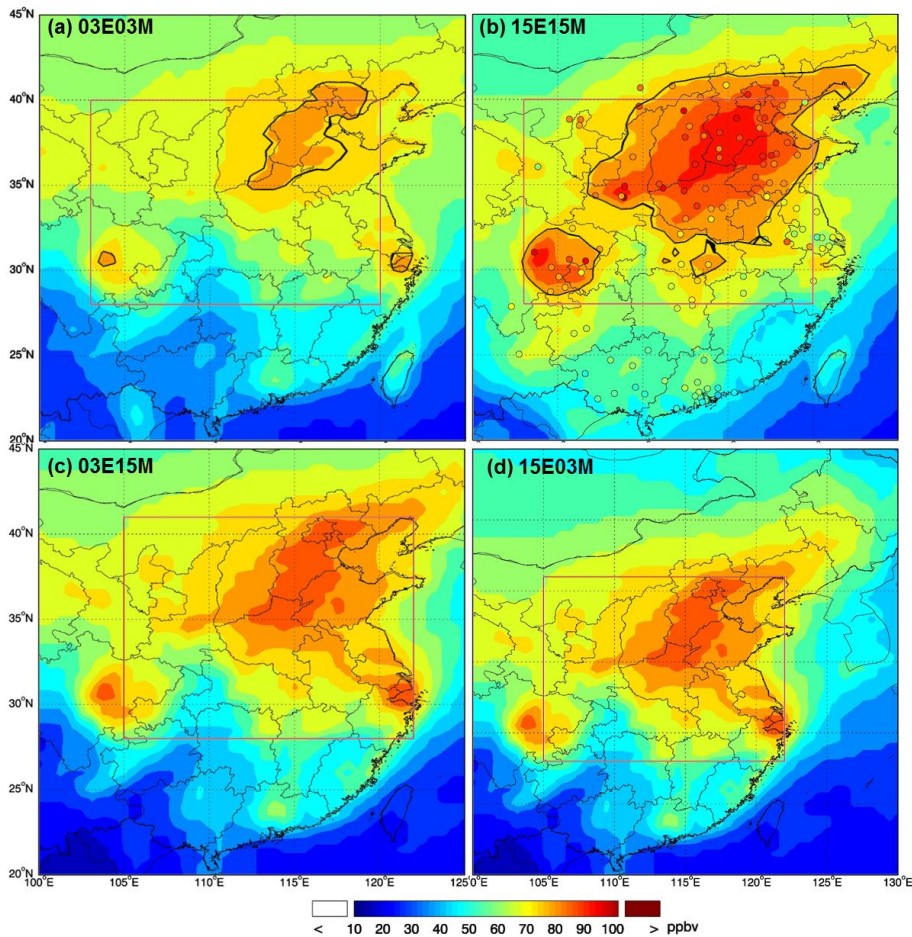

**Figure 2.** Monthly mean spatial distributions of surface MDA8 O$_3$ in July over East China. (a) 03E03M: 2003 standard simulation; (b) 15E15M: 2015 standard simulation; (c) 03E15M: 2003 emission + 2015 meteorology and (d) 15E03M: 2015 emission + 2003 meteorology. Black contours in (a) and (b) indicate the regions with MDA8 O$_3$ > 75 ppbv. Filled circles in (b) show the observed MDA8 O$_3$ at 115 sites of the network of Chinese National Environmental Monitoring Center. The red rectangle represents the Central Eastern China region (CEC: 103 E-120 E, 28 N-40 N).



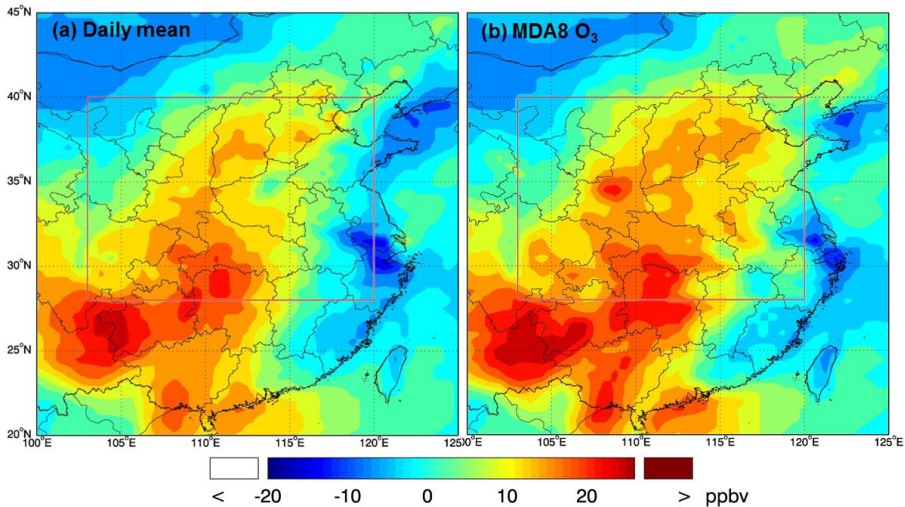

**Figure 3.** Differences in monthly mean surface $O_3$ in July of 2003 and 2015 (2015-2003) for daily mean $O_3$ (a) and MDA8 $O_3$ (b) simulated by 2003 and 2015 standard simulations.

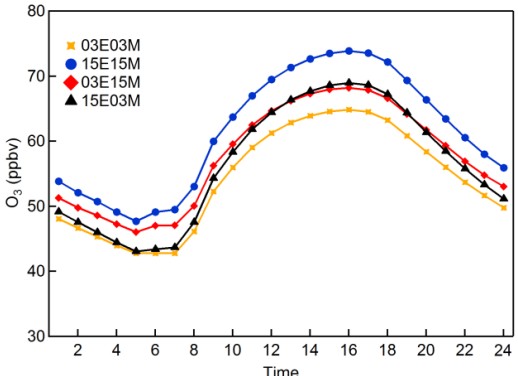

**Figure 4.** Averaged diurnal variations of surface $O_3$ over CEC derived from four modelled results.

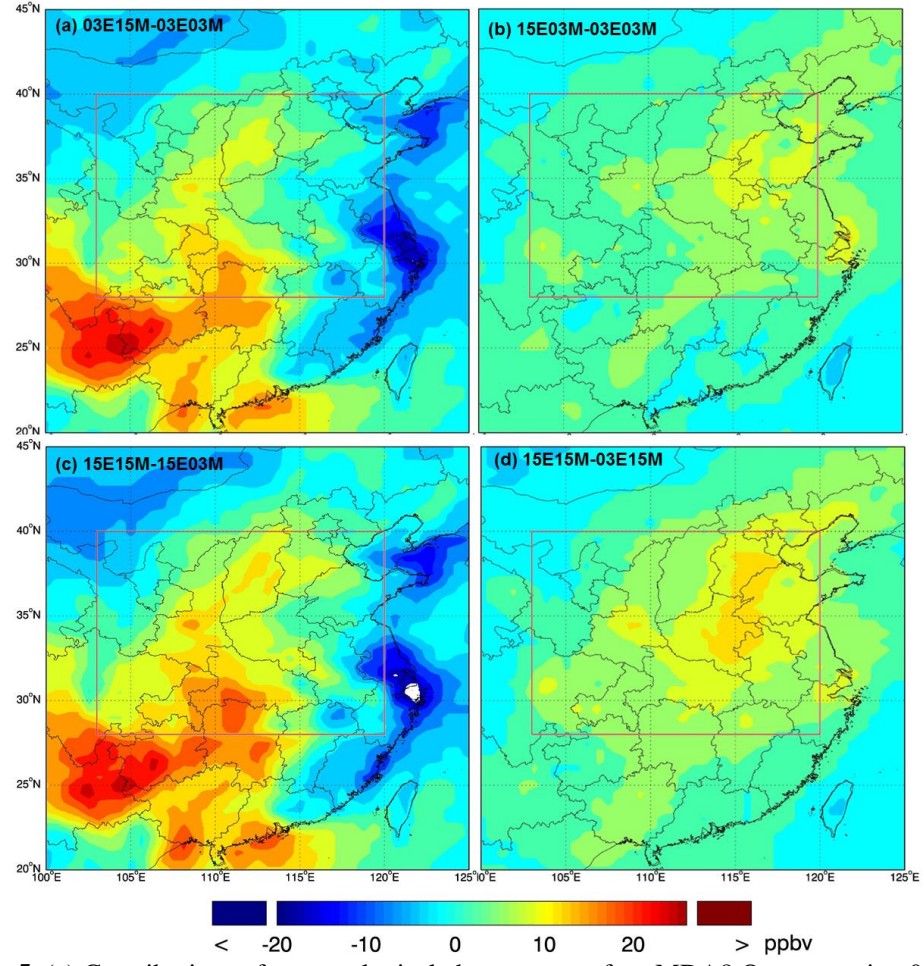

**Figure 5.** (a) Contributions of meteorological changes to surface MDA8 $O_3$, comparing 03E15M and 03E03M (2003 standard) simulations; (b) Contributions of emission changes to surface MDA8 $O_3$, comparing 15E03M and 03E03M (2003 standard) simulations; (c) Contributions of meteorological changes to surface MDA8 $O_3$, comparing 15E15M (2015 standard) and 15E03M simulations; (d) Contributions of emission changes to surface MDA8 $O_3$, comparing 15E15M (2015 standard) and 03E15M simulations.





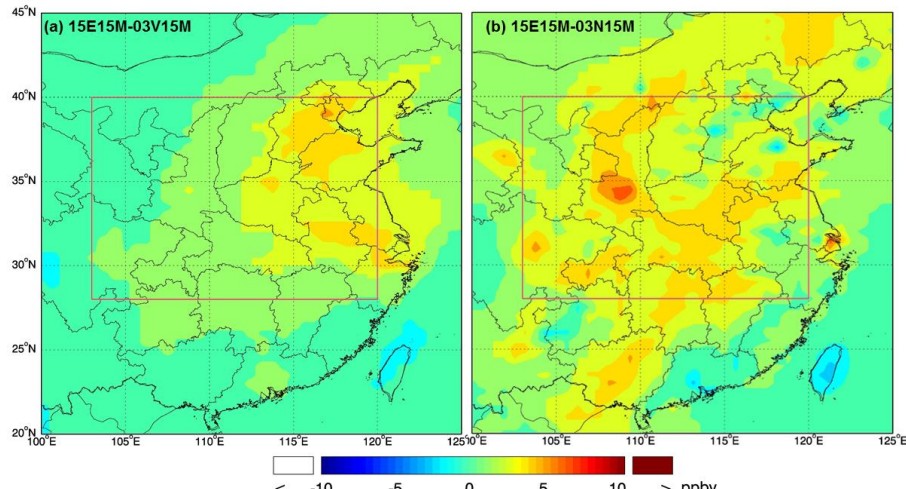

**Figure 6.** Effects of anthropogenic NMVOCs (a) and NOx (b) emission changes on surface MDA8 $O_3$ concentrations between 2003 and 2015 when other emissions and meteorological parameters are fixed at 2015 levels.