# Peer review of "Impacts of meteorology and emissions on summertime surface ozone increases over Central Eastern China between 2003 and 2015"

_Atmospheric Chemistry and Physics, 2018_

## Referee Comment (RC1) · Anonymous Referee #1 · 12 Sep 2018

The manuscript titled "Impacts of meteorology and emissions on surface ozone increases over Central Eastern China between 2003 and 2015" present a very interesting and useful model study showing an increase in surface ozone over Central Eastern China (CEC) between July 2003 and July 2015 which is in agreement with recent studies (Xu et al., 2016, 018; Lu et al., 2018; Gaudel et al., 2018).

According to this present study, emission changes have a higher impact on the Maximum Daily 8-h Average of ozone (MDA8 ozone) than the meteorological changes. However the meteorological changes would better explain larger ozone increases (delta MDA8 ozone >= 10 ppbv between July 2003 and July 2015) than the emission changes. By this latter result, the authors would like to point out that, the long-range transport of ozone and its precursors from neighboring areas should be taken into account, for air pollution control.

The manuscript is well written and the quality of the text and its structure is very much appreciated. However I suggest major revisions, especially regarding one main conclusion of the manuscript that would need more evidence. Indeed, the impact of the transboundary transport on the surface $O_3$ above CEC that would explain an increase of surface $O_3$ between July 2003 and July 2015 is not very much convincing as it is written now.

You can find below general comments followed by more specific comments.

General comments:

1) I would suggest the authors to cite the following recent studies to put even better in perspective the increase of surface ozone between July 2003 and July 2015 using long-term time series:

Xu et al. (2016, 2018), Lu et al. (2018) and Gaudel et al. (2018)

Xu, W., Lin, W., Xu, X., Tang, J., Huang, J., Wu, H. and Zhang, X., 2016. Long-term trends of surface ozone and its influencing factors at the Mt Waliguan GAW station, China–Part 1: Overall trends and characteristics. *Atmospheric Chemistry and Physics*, *16*(10), pp.6191-6205.

Xu, W., Xu, X., Lin, M., Lin, W., Tarasick, D., Tang, J., Ma, J. and Zheng, X., 2018. Long-term trends of surface ozone and its influencing factors at the Mt Waliguan GAW station, China-Part 2: The roles of anthropogenic emissions and climate variability. *Atmospheric Chemistry & Physics*, *18*(2).

Lu, X., Hong, J., Zhang, L., Cooper, O.R., Schultz, M., Xu, X., Wang, T., Gao, M., Zhao, Y. and Zhang, Y., 2018. Severe surface ozone pollution in China: a global perspective. *Environmental Science & Technology Letters*.

Gaudel, A., Cooper, O.R., Ancellet, G., Barret, B., Boynard, A., Burrows, J.P., Clerbaux, C., Coheur, P.F., Cuesta, J., Cuevas Agulló, E. and Doniki, S., 2018. Tropospheric Ozone Assessment Report: Present-day distribution and trends of tropospheric ozone relevant to climate and global atmospheric chemistry model evaluation.

2) I find hard to understand why the authors have chosen the two periods July 2003 and July 2015. Could the authors explain more this choice?

According to the Tropospheric Ozone Assessment Report database (https://join.fz-juelich.de) and seasonal cycles studies (e.g. Sun et al., 2016), the month with higher ozone for most of the sites above CEC would be June. Why would the authors choose July?

In addition, according to Figure 1 of the present manuscript, the observations of surface ozone are available mostly in 2004, why would the authors choose 2003 instead of 2004?

It should be clarified in the text.

3) I didn't find a strong argument supporting the impact of transboundary transport on the surface $O_3$ above CEC that would explain the increase of surface $O_3$ between July 2003 and July 2015. The authors should add more evidence or be clearer in their analysis.

Specific comments:
Title:
I would suggest to add "July" in the title.

Abstract:
Line 23 p.1: Change "The increase in regional averaged $O_3$ resulting from…" to "The increase in averaged $O_3$ in the CEC region resulting from…"

Sections 2 to 6:
Line 19 p.4: The authors are using the global chemical transport model GEOS-Chem v11-01 but the current version reported on the website cited in the manuscript is v11-02. Would the use of v11-02 instead of v11-01 change the results? Could the authors add a word about it in the text?

Line 28 p.4: Could the authors say how long they estimate the spin-up? How many month?

Line 17-20 p.6: I would suggest to change "The contribution of anthropogenic NOx (NMVOCs) emission changes can be calculated by the difference between the 2015 standard simulation and 03N15M (03V15M), defined as 15E15M-03N15M (15E15M -03V15M). " to "The contribution of anthropogenic NOx and NMVOCs emission changes separately can be calculated by the difference between 15E15M (the 2015 standard simulation) and 03N15M (the 2003 NOx emission simulation), and between 15E15M and the 2003 NMVOCs emission simulation (03V15M)."

Line 25 p.6: It is very useful to cite the website link of the Chinese Data Center but unfortunately, there is no English version. Do the authors know whether it is planned to implement the English version? If yes, could the authors say a word about it?

Line 27 p.6: The authors used the word "background sites". If the authors refer to observed ozone at sites which are not influenced by recent, locally emitted or produced anthropogenic pollution, they should use the word "baseline" instead for consistency purposes (Cooper et al., 2014; Dentener et al., 2011)
Cooper, OR, Parrish, DD, Ziemke, J, Balashov, NV, Cupeiro, M, Galbally, IE, Gilge, S, Horowitz, L, Jensen, NR, Lamarque, J-F, Naik, V, Oltmans, SJ, Schwab, J, Shindell, DT, Thompson, AM, Thouret, V, Wang, Y and Zbinden, RM. 2014. Global distribution and trends of tropospheric ozone: An observation-based review. *Elementa: Science of the Anthropocene* **2**. DOI: https://doi.org/10.12952/journal.elementa.000029
Dentener F, Keating T, H Akimoto H, eds. 2011. Hemispheric Transport of Air Pollution 2010: Part A: Ozone and Particulate Matter. *New York: UN*. (Air Pollut. Stud, vol. 17).

Line 4 p.7: Why the authors didn't choose the year 2004 for all the sites? Is July 2003 comparable with July 2004?

Line 12-14 p.7: Is the simulation for 2004 the same as for 2003? Does it start in January 2004? Could the authors make it clearer?

Line 25-30 p.7: Could the authors explain how they chose the "nine representative sites"? Would it be possible to show the diurnal cycle from observations and the simulations for the July month for each site with the standard deviation? The comparison observations/model, day/night would be more straight forward.

Line 3-4 p.8: Does the trend of observed $O_3$ at Mt Tai come from Sun et al. (2016)? The paper should be explicitly cited. Does the simulated increase of about 1.3 ppbv $yr^{-1}$ refer to the same model: nested version of Geos-Chem or GFDL-AM3?
I would suggest showing the time series observations/model in a figure in the supplement material.
Could the authors report the 95% confidence intervals with the trends?

Line 15 p.8: Is the increasing rate calculated from a delta between 2 years looking at one month comparable to the increasing rate calculated from a full time series?
Are the authors sure of the rate of 0.74 ppbv $yr^{-1}$: 74.4 ppbv compare to 65.5 ppbv, 11 years apart would give around 0.8 ppbv $yr^{-1}$, wouldn't it?

Line 29-30 p.8: I am not sure to understand the reason of the choice of focusing on MDA8 $O_3$ and not the daily mean. Could the authors clarify this point?

Line 9 p.9: Please clarify "domain-averaged"?

Line 13 p.10: I would suggest to explain Table 2 earlier in the text when the authors first refer to the Table in section 3.2.

Line 15 p.11: "even if" would suggest that the authors would expect another result for ΔMDA8 $O_3$ greater than 10 ppbv. If it is the case, could the authors say a word about what result they would expect?

Line 4 p.12: Do the authors still mean MDA8 $O_3$ using the words "$O_3$ concentrations"? Please be consistent.

Line 22 p.12: Remove "in".

Line 7-9 p.13: According to Table 4, the larger $O_3$ flux in 15E03M would be 1906 Gg $mon^{-1}$ and not -1232 Gg $mon^{-1}$, is it correct? Could the author explain where -1232 Gg $mon^{-1}$ come from?

Line 14 p.13: Change "excessive $O_3$ production" to "excessive net $O_3$ production"

Line 25-26 p.13: Could the author rephrase the sentence? Half of photochemically formed $O_3$ in the CEC region in July 2003 can not be removed by transport in July 2015 as it is not the same air masses 11 years later.

Line 27-28 p.13: The sentence "[…] the absolute value of $O_3$ transport flux increased by 395 Gg $mon^{-1}$" is confusing. According to Table 4 all fluxes for both horizontal and vertical transport are actually decreasing when comparing simulation 03E03M with 15E15M. This is in agreement with stronger winds in July 2003 than in July 2015 shown in Figure S6.
The sentence needs to be rephrased.

Line 30-31 p.13: "As a result, the increase in $O_3$ concentrations from July 2003 to July 2015 is mainly due to transboundary horizontal transport, vertical transport and photochemical reactions."
Regarding the transboundary horizontal transport, what would be the source regions of $O_3$ that would affect $O_3$ above CEC? Regarding the vertical transport, do the results imply there is an increase in stratospheric intrusion?
According to Figure S6 that shows weaker winds in July 2015 than in July 2003, the transboundary horizontal transport doesn't seem to be a major process that would explain an increase in surface $O_3$ above CEC. Indeed, from Table 4, the local photochemical processes rather than the transport processes seem to be leading the increase in $O_3$. The slow changes in time of the dry deposition process could also explain the increase in surface $O_3$ because it cannot compensate the increase in net photochemical production of $O_3$.

Conclusions:
Line 4 p.14: Would it be possible to put the numbers in perspective with long-term time series study above individual sites such as Mount Waliguan, Mount Tai, Shangdiazi?

Line 14-15 p.14: The terms "transport" can refer to the winds, which are part of the "meteorological" condition. Could the authors explicitly use the words "winds", "humidity" and "temperature"? That would help the reader.

Line 24-25 p.14: To my opinion, the manuscript does not show a strong argument to support the theory of the impact of transboundary transport on surface $O_3$ above CEC. Could the authors be clearer and bring more evidence?

---

## Referee Comment (RC2) · Anonymous Referee #2 · 19 Sep 2018

This manuscript aims to quantify the contributions of emission changes and meteorological conditions on surface ozone changes over Central Eastern China between July 2003 and 2015. An ensemble of simulations using the GEOS-Chem model were conducted to diagnose the impacts of meteorology, anthropogenic and natural emissions. The results show comparable and spatially different contributions from emissions and meteorology on surface ozone changes between the two months, and further point out the importance of chemical production and pollution transport on surface ozone over Central Eastern China. The manuscript is generally well written and fits the scope of ACP. The results are valuable for better understanding the ozone pollution over Eastern China. I recommend publish on ACP but after the following

comments being addressed.

**Main Comments**

1) This study focused on surface ozone changes between July 2003 and July 2015, yet it is not clear why the two particular years (2003 and 2015) were selected. Why not analyze other years, for example, 2014, as the Chinese anthropogenic emissions in the model were based on 2014? Were meteorological conditions in the two years distinctly different from each other, in order to emphasize the impact of meteorology as analysed in this study? Please clarify.

2) Since only two months were analyzed in the study, it needs to be careful with the interpretation of the surface ozone changes between the two periods. The manuscript described the ozone changes mainly as a increasing trend and compared it with previous trend observations, e.g., the paragraphs in Sect. 3.1 and Sect. 3.2 (Page 8). It should be well noted here that surface ozone changes between July 2003 and 2015 may largely reflect ozone inter-annual variability, not a trend.

**Specific Comments**

1) Page 7, Line 4:
"but with the 2004 observations for the other four sites". Do you mean there was no observations available for the other four sites in 2003?

2) Page 7, Line 17:
Please describe more how non-urban sites were selected in this study. Based on the population density or any other information? Did the authors select one site for each city?

3) Page 8, Line 28
"This result is very different from the trends over the US, where summertime daytime

O3 increased over the past decades is contrast to the night-time decrease in all seasons (Yan et al., 2018a)." Yan et al. (2018a) showed that US summertime daytime O3 decreased and nighttime O3 increased in the past decade. Please check.

4) Page 10, Line 8
Please provide values of air temperature and relative humidity over CEC on Figure S7. Seen from this Figure, it seems air temperature in July 2015 was higher than that in July 2003.

5) Page 10, Line 15:
What is "gradient analysis"? Please clarify.

6) Page 12, Figure 6:
I suggest add a figure in the Supplement showing the spatial distribution of changes in anthropogenic NMVOCs and NOx emissions between 2003 and 2015. This can provide helpful information to better interpret their resulting changes in surface O3 as shown in Figure 6.

7) Page 13, Line 4-8:
"minus value" should be "negative value". In "15E03M (-1232 Gg mon-1)", where did "-1232" come from? Table 4 shows "-1100"

8) Page 13, Line 27:
"we find that the absolute value of O3 transport flux increased by 395 Gg mon-1 (2015-2003)". This sentence is misleading. The absolute value of O3 transport flux actually decreased in 2015 relative to 2003 due to less export in 2015. Please clarify.

9) Page 14, Line 2:
"Asia nested model" should be "Asian nested model".

10) Page 14, Line 14:
The statement "The transport pattern in July 2015 tends to enhance O3 levels over the central part of CEC" needs some explanation. Is that because the meteorological conditions in July 2015 favoured pollution accumulation and reduced O3 export over CEC and thus enhanced O3 levels there?

11) In the supplement, Figure S6 and S7:
The meteorological fields should be based on "MERRA-2" instead of "the GEOS-Chem results".

---

## Referee Comment (RC3) · Anonymous Referee #3 · 1 Oct 2018

This paper analyzes the individual impacts of meteorological condition and emission on summertime ozone concentration in Central Eastern China based on GEOS-Chem model. This is generally a solid study with reasonable analyzing and discussion of the model results, and the manuscript is well organized. Therefore I would recommend the manuscript being accepted for publish if the following issues could be properly addressed

Major comments: There exists significant inter-annual variability of meteorological conditions in CEC, did the authors chose these two year (2003, 2015) to conduct the simulation due to their representativeness? Additionally, it is known that China's NOx

emission toped around the year 2011. So the impact of anthropogenic emission in 2011 might reach its maximum rather than in 2015. In addition, since the present work only studied one specific month (July), I personally do not think that the results can extrapolate for the whole year. Therefore, the season with concern ought to be specified in the title.

Process analysis is a diagnostic tool to quantitatively provide the relative contributions from different chemical/physical processes, which is suggested to be discussed with Section 3-4 to further support the conclusion, rather than as an isolated section. For instance, ozone concentration changes due to transport and dry deposition processes may be more closely related to the circulation as well as meteorological conditions, while those due to photochemistry can be interpolated by emission change.

Another suggestion is the inclusion of more in-depth analysis on of precursors' response. Specifically, information on how the changes in emission and meteorology influence spatial pattern of NOx and VOC can help better interpolate the model results.

Minor corrections: Section 3.1: Technically, model evaluation should include performance on reproducing meteorology, relevant precursors as well as ozone.

Table 3: What is the region for the emissions, China or global emission? It should be explained in the caption and also in the main text. Since this work mainly focused on ozone in China, I believe the comparisons of emission in China would make more sense.

Table 4: What the values in the parenthesis stand for? Another, it is better to sum up the horizontal and vertical advection into one single term to represent the contribution of transport.

Figure 5: What does the white color in Figure 5 mean?

Page 14 Line 25: What is difference between transboundary and long-distance transport here, and how the authors draw this conclusion based on this work?

Page 2 Line 8: which controls

Page 14 Line 2: "Asia" should be "Asian"
* * *

---

## Author Comment (AC1) · 28 Nov 2018

**Response to Reviewer 1**

*The manuscript titled "Impacts of meteorology and emissions on surface ozone increases over Central Eastern China between 2003 and 2015" present a very interesting and useful model study showing an increase in surface ozone over Central Eastern China (CEC) between July 2003 and July 2015 which is in agreement with recent studies (Xu et al., 2016, 2018; Lu et al., 2018; Gaudel et al., 2018).*

*According to this present study, emission changes have a higher impact on the Maximum Daily 8-h Average of ozone (MDA8 ozone) than the meteorological changes. However the meteorological changes would better explain larger ozone increases (delta MDA8 ozone >=10 ppbv between July 2003 and July 2015) than the emission changes. By this latter result, the authors would like to point out that, the long-range transport of ozone and its precursors from neighboring areas should be taken into account, for air pollution control.*

*The manuscript is well written and the quality of the text and its structure is very much appreciated. However I suggest major revisions, especially regarding one main conclusion of the manuscript that would need more evidence. Indeed, the impact of the transboundary transport on the surface $O_3$ above CEC that would explain an increase of surface $O_3$ between July 2003 and July 2015 is not very much convincing as it is written now.*

*You can find below general comments followed by more specific comments.*

**Response:** we thank the reviewer for the helpful comments and suggestions, which are very helpful for improving our original manuscript. Below we address all of these comments and have revised the manuscript accordingly. For clarity, the reviewer's comments are listed below in black italics, whilst our responses and changes in the manuscript are shown in blue and red, respectively.

***General comments:***

*1) I would suggest the authors to cite the following recent studies to put even better in perspective the increase of surface ozone between July 2003 and July 2015 using long-term time series: Xu et al. (2016, 2018), Lu et al. (2018) and Gaudel et al. (2018)*

*Xu, W., Lin, W., Xu, X., Tang, J., Huang, J., Wu, H. and Zhang, X., 2016. Long-term trends of surface ozone and its influencing factors at the Mt Waliguan GAW station, China–Part 1: Overall trends and characteristics. Atmospheric Chemistry and Physics, 16(10), pp.6191-6205.*

*Xu, W., Xu, X., Lin, M., Lin, W., Tarasick, D., Tang, J., Ma, J. and Zheng, X., 2018. Long-term trends of surface ozone and its influencing factors at the Mt Waliguan GAW*

*station, China-Part 2: The roles of anthropogenic emissions and climate variability. Atmospheric Chemistry & Physics, 18(2).*

*Lu, X., Hong, J., Zhang, L., Cooper, O.R., Schultz, M., Xu, X., Wang, T., Gao, M., Zhao, Y. and Zhang, Y., 2018. Severe surface ozone pollution in China: a global perspective. Environmental Science & Technology Letters.*

*Gaudel, A., Cooper, O.R., Ancellet, G., Barret, B., Boynard, A., Burrows, J.P., Clerbaux, C., Coheur, P.F., Cuesta, J., Cuevas Agullá, E. and Doniki, S., 2018. Tropospheric Ozone Assessment Report: Present-day distribution and trends of tropospheric ozone relevant to climate and global atmospheric chemistry model evaluation.*

**Response:** these recent studies have been cited in the revised manuscript.

*2) I find hard to understand why the authors have chosen the two periods July 2003 and July 2015. Could the authors explain more this choice? According to the Tropospheric Ozone Assessment Report database (https://join.fz-juelich.de) and seasonal cycles studies (e.g. Sun et al., 2016), the month with higher ozone for most of the sites above CEC would be June. Why would the authors choose July? In addition, according to Figure 1 of the present manuscript, the observations of surface ozone are available mostly in 2004, why would the authors choose 2003 instead of 2004? It should be clarified in the text.*

**Response:** we are sorry that the original discussion is unclear. We chose 2003 and 2015 for simulation mainly because some recent studies (especially our previous study of Sun et al., 2016) have reported the significant increase of summertime ozone over the CEC region. And the modelling results indeed indicated the significant differences in either meteorology or anthropogenic emissions between these two years.

Yes, June is usually the month with the highest surface ozone concentrations in North China. In the present study, June was not selected for simulation because of the varying crop residue burning activities. The crop residue burning usually lasts from late May to late June in CEC, and these emissions had varied greatly over the past decade, which would introduce large uncertainty to the evaluation of impacts from anthropogenic emissions. Thus, we didn't focus on the $O_3$ change simulations in June.

To further confirm the conclusions drawn from the comparison between 2003 and 2015, we have conducted additional simulations for July 2004 and July 2014. There are little difference in the modelled regional-mean and spatial distributions of MDA8 $O_3$ between 2003 and 2004 as well as between 2014 and 2015. Overall, the modelling results in 2004 and 2014 supported the major conclusions derived from 2003 and 2015.

The following discussions have been added in the revised manuscript and the supporting information to elaborate this issue.

The last Paragraph in Section 1:

"This is a follow-up study of Sun et al. (2016) that found a significant increase of summertime $O_3$ at a regional site in North China from 2003 and 2015. We integrate the global GEOS-Chem model and its Asian nested model to investigate the spatial distributions of surface $O_3$ over the whole CEC region, and to quantify the relative contributions from changes in meteorological and anthropogenic emission between 2003 and 2015."

The first graph in Section 2.1:

"The models are run with the full standard NOx-Ox-hydrocarbon-aerosol tropospheric chemistry (Mao et al., 2013) for January to August of 2003 and 2015, including the spin-up time of six months (January to June) for each simulation, but only the results for July are discussed in this paper. The results of August 2003 and 2015 are discussed in supplementary document to confirm the result of this study. Since the crop residue burning usually lasts from late May to late June over CEC and the emissions had varied greatly over the past decade, which introduces large uncertainty to the evaluation of impacts from anthropogenic emissions (Chen et al., 2017; Wu et al., 2018), we don't focus on the $O_3$ change simulations in June. For comparison, we also conducted model simulations for July 2004 and July 2014, and the results supported the major findings obtained from 2003 and 2015 (see results in the supplement)."

Page 7, Line 15:

"The simulated surface $O_3$ in 2004 was also compared against these observations in Figure S2."

Page 9, Line 17:

"The spatial distributions of MDA8 $O_3$ in July 2004 and 2014 in Figure S8 present similar patterns to July 2003 and 2015. The regional mean MDA8 $O_3$ increased from 67.8±6.2 ppbv in July 2004 to 74.8±9.8 ppbv in July 2014. In addition, the regional mean MDA8 $O_3$ increased from 63.4±4.9 ppbv in August 2003 to 73.8±5.0 ppbv in August 2015 (Figure S9). These results are comparable to those derived from the comparison between July 2003 and July 2015. The detail description is provided in the supplement."

Supplementary document:

**1.  The model simulated monthly-mean MDA8 $O_3$ in July 2004 and July 2014**

To further confirm the conclusions drawn from the comparison between 2003 and 2015, we also conducted model simulations for July 2004 and July 2014. Figure S2 shows the comparison of observed versus simulated monthly-mean surface $O_3$ levels at six rural sites in July 2004. The model captures well the observed $O_3$ concentrations at Mt Tai, SDZ, and Mt Hua, with only minor bias (-1.6–4.0 ppbv). In comparison, the model tends to overestimate

the $O_3$ levels at Mt Huang, Lin'an and Hok Tsui. Figure S8 shows similar spatial distributions of MDA8 $O_3$ over CEC in July 2003, 2004, 2014 and 2015. In July 2004, the regions with MDA8 $O_3$ >75 ppbv moved to the south of North China Plain compared to July 2003, mostly due to the different atmospheric circulation patterns. The regional mean MDA8 $O_3$ in July 2004 is 67.8±6.2 ppbv, a little higher than that in July 2003 (65.5±7.9). The regional mean MDA8 $O_3$ in July 2014 is 74.8±9.8 ppbv, which is comparable to that in July 2015 (74.4±8.7 ppbv). We can find the significant increases of MDA8 $O_3$ from 2004 to 2014 as well as from 2003 to 2015. The different concentrations and spatial distributions of $O_3$ between 2003 and 2004 (as well as between 2014 and 2015) should be mostly due to the inter-annual variability in meteorological conditions. Overall, the modelling results in 2004 and 2014 supported the major conclusions derived from 2003 and 2015.

**2.  The model simulated monthly-mean MDA8 $O_3$ in August 2003 and August 2015**

Figure S9 shows the spatial distributions of MDA8 $O_3$ in August 2003 and August 2015. In August 2003, there is no region with monthly-mean MDA8 $O_3$ >75 ppbv. In August 2015, the region with monthly-mean MDA8 $O_3$ >75 ppbv is comparable to that in July 2015, but the $O_3$ levels are generally smaller than those in July 2015. The regional mean MDA8 $O_3$ in August 2003 and 2015 are 63.4±4.9 and 73.8±5.0 ppbv, yielding an increase of 10.4 ppbv. The regional mean MDA8 $O_3$ during July-August of 2003 and 2015 are 64.5±6.4 and 74.1±6.8 ppbv, giving a comparable increase of 9.6 ppbv from 2003 to 2015. The difference in the regional mean MDA8 $O_3$ between July and August of 2003 is 2.1 ppbv, a little higher than that between July and August in 2015 (0.6 ppbv). Such levels are much lower than the difference between 2003 and 2015. Overall, the modelling results in August of 2003 and 2015 supported the major conclusions derived from July 2003 and July 2015.

[Figure]

**Figure S8.** Monthly-mean spatial distributions of surface MDA8 $O_3$ in July over East China. (a) 03E03M: 2003 standard simulation; (b) 04E04M: 2004 standard simulation; (c) 14E14M: 2014 standard simulation and (d) 15E15M: 2015 standard simulation. Black contours indicate the regions with MDA8 $O_3$ > 75 ppbv. Filled circles in (d) show the observed MDA8 $O_3$ at 115 sites of the network of Chinese National Environmental Monitoring Center in July 2015. The red rectangle represents the Central Eastern China region (CEC: 103 °E-120 °E, 28 °N-40 °N).

[Figure]

**Figure S9.** Monthly-mean spatial distributions of surface MDA8 $O_3$ in August over East China. (a) 03E03M: 2003 standard simulation and (b) 15E15M: 2015 standard simulation. Black contours indicate the regions with MDA8 $O_3$ > 75 ppbv. The red rectangle represents the Central Eastern China region (CEC: 103 °E-120 °E, 28 °N-40 °N).

*3) I didn't find a strong argument supporting the impact of transboundary transport on the surface $O_3$ above CEC that would explain the increase of surface $O_3$ between July 2003 and July 2015. The authors should add more evidence or be clearer in their analysis.*

**Response:** we agree with the reviewer. The phrase of "transboundary transport" may be not appropriate, and "transport" should be better here. As indicated by the budget analysis, the transport pattern in July 2015 tended to export less $O_3$-laden air masses than in 2003, which means that more $O_3$ were accumulated within CEC. Since the $O_3$ levels in July 2015 is higher than that in July 2003, if the transport pattern in 2015 was the same as 2003, it would transport much more $O_3$ air masses from the CEC. As well, the difference in transport pattern also led to the different spatial distributions of surface $O_3$ over the CEC region between 2003 and 2015. The following discussions have been added in the revised manuscript to clarify this issue.

"We found that in July 2015, the wind speeds over southern and eastern boundaries of CEC were much lower than that in July 2003 (Figure S13), leading to much lower $O_3$ flux across these two boundaries. The low $O_3$ over southern CEC in July 2003 was mainly due to the strong south-westerly wind, decreasing $O_3$ levels in this area."

"In July 2003, the air flows into CEC through the south boundary, and then out across the other three boundaries. In contrast, the air masses flow into this area across the east boundary in July 2015, and then out across the left three boundaries. The larger $O_3$ flux from each boundary in July 2003 is due to stronger winds. Compared to the 03E03M simulation (-897 Gg mon$^{-1}$; negative value means export of $O_3$ from this region), 03E15M shows a much lower $O_3$ flux (-401 Gg mon$^{-1}$), indicating that weather conditions in 2015 play a more important role in pollutant accumulation, which is consistent with our analysis in Section 4."

*Specific comments:*

*1. Title:*

*I would suggest to add "July" in the title.*

**Response:** the title has been revised as follows. Note that the analysis in the present study covers July and August, and June was not considered due to the variation in biomass burning.

"Impacts of meteorology and emissions on summertime surface ozone increases over Central Eastern China between 2003 and 2015"

*2. Abstract:*

*Line 23 p.1: Change "The increase in regional averaged $O_3$ resulting from…" to "The increase in averaged $O_3$ in the CEC region resulting from…"*

**Response:** changed

*3. Sections 2 to 6:*

*Line 19 p.4: The authors are using the global chemical transport model GEOS-Chem v11-01 but the current version reported on the website cited in the manuscript is v11-02. Would the use of v11-02 instead of v11-01 change the results? Could the authors add a word about it in the text?*

**Response:** we used the version of v11-01 in this study. The web link has been modified as follows.

http://wiki.seas.harvard.edu/geos-chem/index.php/GEOS-Chem_v11-01#v11-01_public_release.

*4. Line 28 p.4: Could the authors say how long they estimate the spin-up? How many month?*

**Response:** the following statements have been added in the revise manuscript to clarify this.

"The models are run with the full standard NOx-Ox-hydrocarbon-aerosol tropospheric chemistry (Mao et al., 2013) for January to August of 2003 and 2015, including the spin-up

time of six months (January to June) for each simulation, but only the results for July are discussed in this paper."

5. *Line 17-20 p.6: I would suggest to change "The contribution of anthropogenic NOx (NMVOCs) emission changes can be calculated by the difference between the 2015 standard simulation and 03N15M (03V15M), defined as 15E15M-03N15M (15E15M -03V15M). " to "The contribution of anthropogenic NOx and NMVOCs emission changes separately can be calculated by the difference between 15E15M (the 2015 standard simulation) and 03N15M (the 2003 NOx emission simulation), and between 15E15M and the 2003 NMVOCs emission simulation (03V15M)."*

**Response:** changed.

6. *Line 25 p.6: It is very useful to cite the website link of the Chinese Data Center but unfortunately, there is no English version. Do the authors know whether it is planned to implement the English version? If yes, could the authors say a word about it?*

**Response:** we found the English version but it does not provide the data now. It may be used in the future. We have added the website of the English version in the revised manuscript.

http://datacenter.mee.gov.cn/aqiweb2/getAirQualityDailyEn (in English)

http://datacenter.mee.gov.cn/websjzx/queryIndex.vm (in Chinese)

7. *Line 27 p.6: The authors used the word "background sites". If the authors refer to observed ozone at sites which are not influenced by recent, locally emitted or produced anthropogenic pollution, they should use the word "baseline" instead for consistency purposes (Cooper et al., 2014; Dentener et al., 2011)*

*Cooper, OR, Parrish, DD, Ziemke, J, Balashov, NV, Cupeiro, M, Galbally, IE, Gilge, S, Horowitz, L, Jensen, NR, Lamarque, J-F, Naik, V, Oltmans, SJ, Schwab, J, Shindell, DT, Thompson, AM, Thouret, V, Wang, Y and Zbinden, RM. 2014. Global distribution and trends of tropospheric ozone: An observation-based review. Elementa: Science of the Anthropocene **2**. DOI: https://doi. org/10.12952/journal.elementa.000029*

*Dentener F, Keating T, H Akimoto H, eds. 2011. Hemispheric Transport of Air Pollution 2010: Part A: Ozone and Particulate Matter. New York: UN. (Air Pollut. Stud, vol. 17).*

**Response:** the "background sites" have been changed to "baseline sites" in the revised manuscript.

8. *Line 4 p.7: Why the authors didn't choose the year 2004 for all the sites? Is July 2003 comparable with July 2004?*

**Response:** the data we have for Mt. Tai and Hok Tsui were in 2003, and the data for the

other sites were only available in 2004 and were taken from literatures. We have simulated the surface $O_3$ in July 2004. The comparison between simulated and observed $O_3$ data in July 2004 is shown in Figure S2. We found the simulated $O_3$ concentrations in July 2003 are comparable to those in July 2004. The original statement has been changed as follows in the revised manuscript.

"We compare the simulated surface $O_3$ concentrations with the 2003 observations for Mt Tai and Hok Tsui but with the 2004 observations for the other four sites (Figure 1(a)). The simulated surface $O_3$ in 2004 was also compared against these observations in Figure S2."

9.  *Line 12-14 p.7: Is the simulation for 2004 the same as for 2003? Does it start in January 2004? Could the authors make it clearer?*

**Response:** yes, the modeling set-up was the same between these simulations. It started from January 2004.

10. *Line 25-30 p.7: Could the authors explain how they chose the "nine representative sites"? Would it be possible to show the diurnal cycle from observations and the simulations for the July month for each site with the standard deviation? The comparison observations/model, day/night would be more straightforward.*

**Response:** we chose non-urban sites to represent the $O_3$ concentrations in major cities over CEC. In general, the selected non-urban sites are sub-urban or rural sites which are far away from the urban and industrialized areas. We have compared the diurnal cycles of $O_3$ (and CO and NOx) from observations and simulations. The following discussions have been added in the revised manuscript and supplements.

"To avoid the influence of local emission, photochemical and deposition processes in small-scales of urban area, we selected one non-urban site to represent the $O_3$ concentrations of each city over CEC. In general, the selected non-urban sites are sub-urban or rural sites which are far away from the urban and industrialized areas. For cities where no non-urban sites are available, we chose the stations that are least affected by local pollution (i.e., sites relatively far away from roads, factories, power plants, etc.)."

"Time series and diurnal variations of hourly $O_3$ concentrations from the model and observations at Mt Tai in 2003 and nine representative sites in 2015 are compared in Figures S3, S4 and S5, respectively. The nine observation sites are carefully selected to be far away from urban areas in the capital cities of nine provinces and municipalities, including Beijing, Tianjin, Ji'nan, Taiyuan, Zhengzhou, Wuhan, Chongqing, Changsha, and Nanjing. The model reproduces the time series of $O_3$ with a normalized mean bias of 4% at Mt Tai. The overestimation of $O_3$ concentrations in the afternoon is likely to be due to the overestimated precursor emissions in the model. For the nine sites, the model captures most day-to-day

variability and diurnal variations (Figures S4 and S5)."

*11. Line 3-4 p.8: Does the trend of observed O$_3$ at Mt Tai come from Sun et al. (2016)? The paper should be explicitly cited. Does the simulated increase of about 1.3 ppbv yr$^{-1}$ refer to the same model: nested version of Geos-Chem or GFDL-AM3? I would suggest showing the time series observations/model in a figure in the supplement material. Could the authors report the 95% confidence intervals with the trends?*

**Response:** the original statement is confusing. Yes, the trend of observed O$_3$ at Mt. Tai was taken from Sun et al. (2016), and the simulated increase of ~1.3 ppbv yr$^{-1}$ was calculated from the Geos-Chem model in the present study. The use of "trend" should be not appropriate as we only performed simulations for two years. The original statement has been modified as follows in the revised manuscript.

"In addition, the observed results of Sun et al. (2016) reported the MDA8 O$_3$ at Mt Tai increased from 75.9±15.9 to 102.1±28.1 ppbv in July-August from 2003 to 2015, which is higher than the simulated result in this study (i.e., from 71.1±10.0 ppbv in July 2003 to 90.4±18.5 ppbv in July 2015). Nonetheless, the model captures the significant increase in surface O$_3$ levels over CEC between July 2003 and July 2015."

*12. Line 15 p.8: Is the increasing rate calculated from a delta between 2 years looking at one month comparable to the increasing rate calculated from a full time series? Are the authors sure of the rate of 0.74 ppbv yr$^{-1}$: 74.4 ppbv compare to 65.5 ppbv, 11 years apart would give around 0.8 ppbv yr$^{-1}$, wouldn't it?*

**Response:** we agree with the reviewer, and the original descriptions have been modified as follows.

"Table 2 shows the monthly mean MDA8 O$_3$ over CEC. The regional mean MDA8 O$_3$ increased from 65.5±7.9 ppbv in July 2003 to 74.4±8.7 ppbv in July 2015, showing an increase of about 8.9±3.9 ppbv in twelve years. According to the limited reports of observed long-term (>10 years) changes of O$_3$ concentrations, we find significant increases of summertime O$_3$ (1–3 ppbv yr$^{-1}$) in the north part (Beijing), east part (Mt Tai) and south part (Lin'an) of CEC over the past two decades (Ding et al., 2008; Ma et al., 2016; Sun et al., 2016; Xu et al., 2008; Zhang et al., 2014b). Our result shows that both daily mean O$_3$ concentration and MDA8 O$_3$ were significantly higher in July 2015 than in July 2003 over most areas of CEC (Figure 3)."

*13. Line 29-30 p.8: I am not sure to understand the reason of the choice of focusing on MDA8 O$_3$ and not the daily mean. Could the authors clarify this point?*

**Response:** MDA8 O$_3$ has been widely used as an indicator to present the O$_3$ pollution levels. And due to the NO titration at nighttime, the O$_3$ concentrations at night may not reflect the actual O$_3$ pollution conditions. So we choose the MDA8 O$_3$ for analyses, rather than the daily

mean. The following statement has been added in the revised manuscript to clarify this.

"Considering that the nighttime $O_3$ is easily titrated by NO and the MDA8 $O_3$ is a good indicator for the overall $O_3$ pollution condition, we focus on the MDA8 $O_3$ changes over CEC between July 2003 and July 2015 instead of daily-mean $O_3$."

*14. Line 9 p.9: Please clarify "domain-averaged"?*

**Response:** it has been changed to "regional mean".

*15. Line 13 p.10: I would suggest to explain Table 2 earlier in the text when the authors first refer to the Table in section 3.2.*

**Response:** we adopt this suggestion and move it to section 3.2.

*16. Line 15 p.11:"even if" would suggest that the authors would expect another result for ΔMDA8 $O_3$ greater than 10 ppbv. If it is the case, could the authors say a word about what result they would expect?*

**Response:** the original statements may be unclear and have been clarified as follows in the revised manuscript.

"It is worth noting that in the polluted regions where MDA8 $O_3$ >75 ppbv, the contribution of emission change increases from 5.0±1.8 ppbv for ΔMDA8 $O_3$ ≥0 ppbv case to 5.2±1.7 ppbv for ΔMDA8 $O_3$ ≥10 ppbv case, whilst the contribution of meteorology change increases from 3.7±3.2 ppbv to 5.0±2.5 ppbv. Even if the ΔMDA8 $O_3$ is greater than 10 ppbv, the $O_3$ increase caused by emission change is still higher than that caused by meteorological change, indicating the dominant effect of emissions on $O_3$ pollution in the highly polluted regions."

*17. Line 4 p.12: Do the authors still mean MDA8 $O_3$ using the words "$O_3$ concentrations"? Please be consistent.*

**Response:** we have changed the "$O_3$ concentration" to "MDA8 $O_3$".

*18. Line 22 p.12: Remove "in".*

**Response:** removed

*19. Line 7-9 p.13: According to Table 4, the larger $O_3$ flux in 15E03M would be 1906 Gg mon$^{-1}$ and not -1232 Gg mon$^{-1}$, is it correct? Could the author explain where -1232 Gg mon$^{-1}$ come from?*

**Response:** it should be -1100 Gg mon$^{-1}$ and has been corrected in the revised manuscript.

*20. Line 14 p.13: Change "excessive $O_3$ production" to "excessive net $O_3$ production"*

**Response:** changed.

*21. Line 25-26 p.13: Could the author rephrase the sentence? Half of photochemically*

*formed $O_3$ in the CEC region in July 2003 cannot be removed by transport in July 2015 as it is not the same air masses 11 years later.*

**Response:** agree. This statement has been rephrased as follows.

"In July 2003, about half of the net photochemically formed $O_3$ in the CEC region was removed by transport (897 out of 1629 Gg mon$^{-1}$). In comparison, only 1/4 of the net photochemically produced $O_3$ (502 out of 2158 Gg mon$^{-1}$) was transported out of CEC in July 2015."

*22. Line 27-28 p.13: The sentence "[...] the absolute value of $O_3$ transport flux increased by 395 Gg mon-1" is confusing. According to Table 4 all fluxes for both horizontal and vertical transport are actually decreasing when comparing simulation 03E03M with 15E15M. This is in agreement with stronger winds in July 2003 than in July 2015 shown in Figure S6. The sentence needs to be rephrased.*

**Response:** the original sentence has been rephrased as follows.

"Comparing the results of the 2003 and 2015 standard simulations (15E15M-03E03M), we find less $O_3$ export from CEC in 2015, which means about 395 Gg mon$^{-1}$ (2015-2003) $O_3$ was accumulated in this region."

*23. Line 30-31 p.13:*

*"As a result, the increase in $O_3$ concentrations from July 2003 to July 2015 is mainly due to transboundary horizontal transport, vertical transport and photochemical reactions."*

*Regarding the transboundary horizontal transport, what would be the source regions of $O_3$ that would affect $O_3$ above CEC? Regarding the vertical transport, do the results imply there is an increase in stratospheric intrusion?*

*According to Figure S6 that shows weaker winds in July 2015 than in July 2003, the transboundary horizontal transport doesn't seem to be a major process that would explain an increase in surface $O_3$ above CEC. Indeed, from Table 4, the local photochemical processes rather than the transport processes seem to be leading the increase in $O_3$. The slow changes in time of the dry deposition process could also explain the increase in surface $O_3$ because it cannot compensate the increase in net photochemical production of $O_3$.*

**Response:** the original statement is somewhat misleading. The increase in $O_3$ concentrations from July 2003 to July 2015 should be due to the enhanced photochemical production (mainly due to the increased emissions) and the weakened export (due to the meteorological conditions). This statement has been revised as follows in the revised manuscript.

"As a result, the increase in $O_3$ concentrations from July 2003 to July 2015 should be due to the enhanced photochemical production (mainly due to the increased emissions) and the

weakened export (due to the meteorological conditions)."

*Conclusions:*

*24. Line 4 p.14: Would it be possible to put the numbers in perspective with long-term time series study above individual sites such as Mount Waliguan, Mount Tai, Shangdiazi?*

**Response:** the results from these long-term time series study have been discussed in the main context, and were not listed again in the conclusion section.

*25. Line 14-15 p.14: The terms "transport" can refer to the winds, which are part of the "meteorological" condition. Could the authors explicitly use the words "winds", "humidity" and "temperature"? That would help the reader.*

**Response:** the original statement has been modified as follows.

"The meteorological conditions (mostly due to wind patterns) in July 2015 tended to accumulate pollution and reduced $O_3$ export over the central part of CEC and thus enhanced $O_3$ levels there. Air temperature and relative humidity does not promote the $O_3$ production in July 2015."

*Line 24-25 p.14: To my opinion, the manuscript does not show a strong argument to support the theory of the impact of transboundary transport on surface $O_3$ above CEC. Could the authors be clearer and bring more evidence?*

**Response:** the phrase of "transboundary transport" may be misleading. From the budget analysis, we found that large-scale regional transport is an important contributor to the spatial distributions and inter-annual variations of surface $O_3$ over the CEC region. The original statement has been modified as follows.

"Transport issues in local $O_3$ control strategies should go beyond transport from neighbouring areas (e.g., cities) and account for the long-distance transport (e.g., across provinces)."

---

## Author Comment (AC2) · 28 Nov 2018

**Response to Referee 2**

*This manuscript aims to quantify the contributions of emission changes and meteorological conditions on surface ozone changes over Central Eastern China between July 2003 and 2015. An ensemble of simulations using the GEOS-Chem model were conducted to diagnose the impacts of meteorology, anthropogenic and natural emissions. The results show comparable and spatially different contributions from emissions and meteorology on surface ozone changes between the two months, and further point out the importance of chemical production and pollution transport on surface ozone over Central Eastern China. The manuscript is generally well written and fits the scope of ACP. The results are valuable for better understanding the ozone pollution over Eastern China. I recommend publish on ACP but after the following comments being addressed.*

**Response:** we thank the referee for the positive comments and helpful suggestions. We have addressed all of these comments in the revised manuscript, as detailed below in the responses to individual comments. For clarify, the referee's comments are listed below in black italics, while our responses and changes in the manuscript are shown in blue and red, respectively.

*Main Comments*

*1) This study focused on surface ozone changes between July 2003 and July 2015, yet it is not clear why the two particular years (2003 and 2015) were selected. Why not analyze other years, for example, 2014, as the Chinese anthropogenic emissions in the model were based on 2014? Were meteorological conditions in the two years distinctly different from each other, in order to emphasize the impact of meteorology as analyzed in this study? Please clarify.*

**Response:** we are sorry that the original discussion is unclear. We chose 2003 and 2015 for simulation mainly because some recent studies (especially our previous study of Sun et al., 2016) have reported the significant increase of summertime ozone over the CEC region. And the modelling results indeed indicated the significant differences in either meteorology or anthropogenic emissions between these two years. To further confirm the conclusions drawn from the comparison between 2003 and 2015, we have conducted additional simulations for July 2004 and July 2014. There are little difference in the modelled regional-mean and spatial distributions of MDA8 $O_3$ between 2003 and 2004 as well as between 2014 and 2015. The difference of MDA8 $O_3$ between 2003 and 2014 (or 2004 and 2015) is much higher than the difference between 2003 and 2004 (or 2014 and 2015). This means that the impact of inter-annual variability should be smaller than the decadal change in meteorology and emissions. The following discussions have been added in the revised manuscript and supporting information to elaborate this issue.

"This is a follow-up study of Sun et al. (2016) that found a significant increase of summertime $O_3$ at a regional site in North China from 2003 and 2015. We integrate the global GEOS-Chem model and its Asian nested model to investigate the spatial distributions of surface $O_3$ over the whole CEC region, and to quantify the relative contributions from changes in meteorological and anthropogenic emission between 2003 and 2015."

"For comparison, we also conducted additional model simulations for 2004 and 2014, and the results supported the major findings obtained from 2003 and 2015 (see results in the supplement)."

"In July 2004, the regions with MDA8 $O_3$ > 75 ppb moved to south of North China Plain compared to July 2003, mostly due to the different atmospheric circulation patterns. The regional-mean MDA8 $O_3$ in July 2004 is 67.8±6.2 ppbv, a little higher than that in July 2003 (65.5±7.9). In comparison, the regional mean MDA8 $O_3$ in July 2014 is 74.8±9.8 ppbv, which is comparable to that in July 2015 (74.4±8.7 ppbv)."

[Figure]

**Figure S8.** Monthly-mean spatial distributions of surface MDA8 $O_3$ in July over East China. (a) 03E03M: 2003 standard simulation; (b) 04E04M: 2004 standard simulation; (c) 14E14M: 2014 standard simulation and (d) 15E15M: 2015 standard simulation. Black contours indicate the regions with MDA8 $O_3$ > 75 ppbv. Filled circles in (d) show the observed MDA8 $O_3$ at 115 sites of the network of Chinese National Environmental Monitoring Center in July 2015. The red rectangle represents the Central Eastern China region (CEC: 103°E-120°E, 28°N-40°N).

*2) Since only two months were analyzed in the study, it needs to be careful with the interpretation of the surface ozone changes between the two periods. The manuscript described the ozone changes mainly as an increasing trend and compared it with previous trend observations, e.g., the paragraphs in Sect. 3.1 and Sect. 3.2 (Page 8). It should be well noted here that surface ozone changes between July 2003 and 2015 may largely reflect ozone inter-annual variability, not a trend.*

**Response:** we totally agree with the reviewer that the $O_3$ changes between two specific years are largely influenced by the inter-annual variability, rather than a trend. We have deleted the

phrase of "trend" in the revised manuscript. Nonetheless, the long-term observations have indeed indicated a significant increasing trend of $O_3$ over CEC in the past decades, and our model simulations in 2003-2004 and 2014-2015 also supported this argument.

**Specific Comments**

*1) Page 7, Line 4:*

*"but with the 2004 observations for the other four sites". Do you mean there was no observations available for the other four sites in 2003?*

**Response:** yes, the observational data at these four sites were obtained in 2004, not in 2003. We have compared the 2004 modelling results with the observation data in 2004. Generally, the simulation results for the six sites in July 2004 are comparable to those in 2003, as shown below.

[Figure]

**Figure S2.** Comparison of observed versus simulated monthly-mean $O_3$ concentrations at the six rural sites in July 2004.

*2) Page 7, Line 17:*

*Please describe more how non-urban sites were selected in this study. Based on the population density or any other information? Did the authors select one site for each city?*

**Response:** the following descriptions have been added in the revised manuscript to clarify this issue.

"To avoid the influence of local emission, photochemical and deposition processes in small-scales of urban area, we selected one non-urban site to represent the $O_3$ concentrations of each city over CEC. In general, the selected non-urban sites are sub-urban or rural sites which are far away from the urban and industrialized areas. For cities where no non-urban sites are available, we chose the stations that are least affected by local pollution (i.e., sites relatively far away from traffic roads, factories, power plants, etc.). As a result, 115

non-urban sites were selected to represent 115 cities in the East China region."

*3) Page 8, Line 28*

*"This result is very different from the trends over the US, where summertime daytime O₃ increased over the past decades is contrast to the night-time decrease in all seasons (Yan et al., 2018a)." Yan et al. (2018a) showed that US summertime daytime O₃ decreased and nighttime O₃ increased in the past decade. Please check.*

**Response:** the original statement has been changed as follows.

"This result is very different from the trends over the US, where summertime daytime O₃ declined over the past decades is contrast to the night-time growth in all seasons (Yan et al., 2018a)"

*4) Page 10, Line 8*

*Please provide values of air temperature and relative humidity over CEC on Figure S7. Seen from this Figure, it seems air temperature in July 2015 was higher than that in July 2003.*

**Response:** we have added the values of air temperature and relative humidity over CEC in the revised manuscript. Air temperature was almost the same between July 2003 and July 2015. Relative humidity in July 2015 was a little lower than that in July 2003.

"Figure S14 shows the simulated spatial distributions of air temperature and relative humidity in July 2003 and July 2015. The simulated air temperatures in 2003 and 2015 are 300.6±3.2 and 300.5±3.2 K, respectively, almost at the same level. The simulated relative humidity in 2003 was 82±10%, a little higher than in 2015 (77±12%). The average net O₃ production over CEC simulated by 03E03M (11.7 ppbv day$^{-1}$) is very close to that simulated by 03E15M (11.9 ppbv day$^{-1}$) (Table 4), suggesting that meteorological factors in 2003 and 2015 did not greatly change O₃ photochemical reactions. Therefore, neither air temperature nor relative humidity plays an important role in explaining the difference in surface O₃ between 2003 and 2015."

*5) Page 10, Line 15:*

*What is "gradient analysis"? Please clarify.*

**Response:** the following descriptions have been added in the revised manuscript to clarify this.

"We performed a gradient analysis, which selected different levels for the difference of MDA8 O₃ (ΔMDA8 O₃) between 2003 and 2015 standard simulations (15E15M-03E03M). The differences in MDA8 O₃ were analysed in four ways: regional mean, ΔMDA8 O₃ ≥0 ppbv, ΔMDA8 O₃ ≥5 ppbv and ΔMDA8 O₃ ≥10 ppbv."

*6) Page 12, Figure 6:*

*I suggest add a figure in the Supplement showing the spatial distribution of changes in anthropogenic NMVOCs and NOx emissions between 2003 and 2015. This can provide helpful information to better interpret their resulting changes in surface O₃ as shown in*

*Figure 6.*

**Response:** thanks for the good suggestion. We have added the spatial distributions of NOx and NMVOC emissions and their changes in the supplement (see below). Some brief discussions have been also added in the revised manuscript.

[Figure]

**Figure S13.** Anthropogenic emissions of NOx in July 2003 (a) and July 2015 (b) and NMVOCs in July 2003 (c) and July 2015 (d). Unit: Mg/Km$^2$/mon for NOx; Mg(C)/Km$^2$/mon for NMVOCs. The red rectangle represents the Central Eastern China region.

[Figure]

**Figure S14.** Spatial distributions of changes in NOx (a) and NMVOCs (b) emissions between July 2003 and July 2015 (2015-2003). Unit: Mg/Km$^2$/mon for NOx; Mg(C)/Km$^2$/mon for NMVOCs. The red rectangle represents the Central Eastern China region.

*7) Page 13, Line 4-8:*

*"minus value" should be "negative value". In "15E03M (-1232 Gg mon-1)", where did*

*"-1232" come from? Table 4 shows "-1100"*

**Response:** these typos have been corrected in the revised manuscript.

*8) Page 13, Line 27:*

*"we find that the absolute value of $O_3$ transport flux increased by 395 Gg mon$^{-1}$ (2015-2003)". This sentence is misleading. The absolute value of $O_3$ transport flux actually decreased in 2015 relative to 2003 due to less export in 2015. Please clarify.*

**Response:** the original statement has been changed as follows in the revised manuscript.

"Comparing the results of the 2003 and 2015 standard simulations (15E15M-03E03M), we find less $O_3$ export from CEC in 2015 than in 2003, which means about 395 Gg mon$^{-1}$ of $O_3$ was accumulated in this region in 2015."

*9) Page 14, Line 2:*

*"Asia nested model" should be "Asian nested model".*

**Response:** changed.

*10) Page 14, Line 14:*

*The statement "The transport pattern in July 2015 tends to enhance $O_3$ levels over the central part of CEC" needs some explanation. Is that because the meteorological conditions in July 2015 favored pollution accumulation and reduced $O_3$ export over CEC and thus enhanced $O_3$ levels there?*

**Response:** the original statement has been changed as follows to clarify this issue.

"The meteorological conditions (especially wind patterns) in July 2015 tended to accumulate pollution and reduced $O_3$ export over the central part of CEC and thus enhanced $O_3$ levels there. Air temperature and relative humidity does not promote the $O_3$ production in July 2015."

*11) In the supplement, Figure S6 and S7:*

*The meteorological fields should be based on "MERRA-2" instead of "the GEOS-Chem results".*

**Response:** changed.

---

## Author Comment (AC3) · 28 Nov 2018

**Response to Referee 3**

*This paper analyzes the individual impacts of meteorological condition and emission on summertime ozone concentration in Central Eastern China based on GEOS-Chem model. This is generally a solid study with reasonable analyzing and discussion of the model results, and the manuscript is well organized. Therefore, I would recommend the manuscript being accepted for publish if the following issues could be properly addressed*

**Response:** we thank the reviewer for the thoughtful review and constructive comments. These comments and suggestions are very helpful for improving our manuscript. We have tried to address all of the referee's comments in the revised manuscript. Below we reply in detail to the individual comments. For clarify, the reviewer's comments are listed in black italics, while our responses and changes are highlighted in blue and red, respectively.

*Major comments: There exists significant inter-annual variability of meteorological conditions in CEC, did the authors chose these two year (2003, 2015) to conduct the simulation due to their representativeness? Additionally, it is known that China's NOx emission toped around the year 2011. So the impact of anthropogenic emission in 2011 might reach its maximum rather than in 2015. In addition, since the present work only studied one specific month (July), I personally do not think that the results can extrapolate for the whole year. Therefore, the season with concern ought to be specified in the title.*

**Response:** we are sorry that the original discussion may be unclear. We chose 2003 and 2015 for simulation mainly because some recent studies have reported the significant increase of summertime ozone over the CEC region (Sun et al., 2016; Ma et al., 2016). Yes, the anthropogenic emissions of NOx have been reduced since 2011, but the NMVOC emissions have continued increasing unabated. Another reason why we didn't choose 2011 for simulation is the lack of observational data for comparison and model validation. The following discussion has been added in the revised manuscript to clearly state the reason for selecting 2003 and 2015.

Page 4 Line 8: "This is a follow-up study of Sun et al. (2016) that found a significant increase of summertime $O_3$ at a regional site in North China from 2003 and 2015. We integrate the global GEOS-Chem model and its Asian nested model to investigate the spatial distributions of surface $O_3$ over the whole CEC region, and to quantify the relative contributions from changes in meteorological and anthropogenic emission between 2003 and 2015."

We agree with the reviewer that the season with concern should be specified in the title. We also performed the modeling analyses for August in the revision process, and the results are similar to those obtained for July. The title has been revised to "Impacts of meteorology and emissions on summertime surface ozone increases over Central Eastern China between 2003

and 2015"

*Process analysis is a diagnostic tool to quantitatively provide the relative contributions from different chemical/physical processes, which is suggested to be discussed with Section 3-4 to further support the conclusion, rather than as an isolated section. For instance, ozone concentration changes due to transport and dry deposition processes may be more closely related to the circulation as well as meteorological conditions, while those due to photochemistry can be interpolated by emission change.*

**Response:** there are indeed some relationships between Sections 3-4 and Section 6, and the budget analysis in Section 6 can quantitatively support the results discussed in Sections 3-4. Nonetheless, we still want to retain the original structure of the manuscript, as we think the current discussions of the impacts of emissions vs. meteorology and transport vs. chemistry in separate sections are also clear enough. In the revised manuscript, we have referred to the results of budget analyses in Section 6 when discussing the relative contributions from emissions and meteorological conditions in Sections 3-4.

*Another suggestion is the inclusion of more in-depth analysis on of precursors' response. Specifically, information on how the changes in emission and meteorology influence spatial pattern of NOx and VOC can help better interpolate the model results.*

**Response:** we totally agree with the reviewer that the analysis of precursors' response could help to better understand the changes of $O_3$. So we examined the spatial distributions of $NO_2$ and NMVOCs for the four modelling scenarios as well as their differences. The following discussions have been added in the revised manuscript and supplementary document.

Page 10, Line 9:

"The spatial distributions of $O_3$ precursors ($NO_2$ and NMVOCs) for the different scenarios and their differences are shown in Figure S10 and S11, which can better explain these results. Detailed description is given in the supplementary document."

Page 12, Line 18:

"The changes of $NO_2$ and NMVOCs also indicate the impact of emission changes larger than that of meteorological change (Figure S10 and S11)."

Supplementary document:

**Spatial distributions of the modelled $O_3$ precursors over CEC in July 2003 and 2015**

The spatial distributions of the modelled $NO_2$ and NMVOCs over CEC in July 2003 and 2015 and their differences are shown in Figures S10 and S11, respectively. We found that both $NO_2$ and NMVOCs had increased significantly over CEC from July 2003 to July 2015. The spatial distribution is in accordance with the emission inventory in Figure S15, which

shows high levels of NO$_2$ and NMVOCs in the eastern CEC and Sichuan basin. Comparing the results of 03E15M-03E03M and 15E03M-03E03M, we can find that the contribution from the emission change (15E03M-03E03M) is much higher than that from the meteorology change (03E15M-03E03M). This is as expected as the O$_3$ precursors are primary pollutants and should be governed by the anthropogenic emissions. Furthermore, the O$_3$ precursor concentrations over the eastern part of CEC increased much higher than the western part. Overall, the modelling results for NO$_2$ and NMVOCs agree well with the results of O$_3$ in the main context.

[Figure]

**Figure S10**. Monthly-mean spatial distributions of surface NO$_2$ in July over CEC: (a)-(d) and the differences in NO$_2$ concentrations between these simulations: (e)-(h). The red rectangle represents the Central Eastern China region (CEC: 103°E-120°E, 28°N-40°N).

[Figure]

**Figure S11**. Monthly-mean spatial distributions of surface NMVOCs in July over CEC: (a)-(d) and the differences in NMVOC concentrations between these simulations: (e)-(h). The concentrations of NMVOCs include: ALK4 (lumped >=C4 Alkanes), Isoprene, Acetone, Methyl Ethyl Ketone, Acetaldehyde, RCHO (lumped Aldehyde >=C3), PRPE (lumped >=C3 Alkenes), Formaldehyde, Hydroxyacetone and Glycoaldehyde. The red rectangle represents the Central Eastern China region (CEC: 103 E-120 E, 28 N-40 N).

*Minor corrections: Section 3.1: Technically, model evaluation should include performance on reproducing meteorology, relevant precursors as well as ozone.*

**Response:** we have compared the observed diurnal variations of NO$_2$, CO and O$_3$ with the simulation results at the nine sites over CEC in July 2015. Overall, the model captures the observed diurnal variations of these species at these sites. We didn't get the meteorological data from the observation sites, thus we didn't compare the meteorological conditions in this study. The following model validation results and discussions have been added in the revised manuscript and supporting materials.

Page 8, Line 20: "We also compared the simulated diurnal variations of CO and NO$_2$ in the nine cities against the observational data (see Figures S6 and S7). Overall, the model captures most diurnal variations of CO and NO$_2$. The underestimation of CO by the model may be due to the underestimation of emissions and/or the excessive OH (Yan et al., 2014; Young et al., 2013). The large bias in NO$_2$ may be due to the effect of local emissions. Another reason for the discrepancy between observed and modelled NO$_2$ is the overestimation by the measurements based on catalytic conversion of other oxidized nitrogen species to NO (Xu et al., 2013)."

[Figure]

**Figure S5.** Observed and simulated monthly-mean diurnal variations of surface O₃ in July 2015 at representative air quality monitoring stations in nine cities.

[Figure]

**Figure S6.** Observed and simulated monthly-mean diurnal variations of surface CO in July 2015 at representative air quality monitoring stations in nine cities.

[Figure]

**Figure S7.** Observed and simulated monthly-mean diurnal variations of surface NO₂ in July 2015 at representative air quality monitoring stations in nine cities.

*Table 3: What is the region for the emissions, China or global emission? It should be explained in the caption and also in the main text. Since this work mainly focused on ozone in China, I believe the comparisons of emission in China would make more sense.*

**Response:** the emission region is the Central Eastern China. We have stated the region in the revised Table caption and the main text.

*Table 4: What the values in the parenthesis stand for? Another, it is better to sum up the horizontal and vertical advection into one single term to represent the contribution of*

*transport.*

**Response:** The values in the parenthesis stand for the amounts of photochemical production and loss in unit of ppbv day$^{-1}$. The "total transport" term represents the sum of horizontal and vertical advection, and thus represents the contribution of transport. We have elaborated this in the revised manuscript.

*Figure 5: What does the white color in Figure 5 mean?*

**Response:** We have modified the color bar of Figure 5, and the revised figure is as follows.

[Figure]

**Figure 5.** (a) Contributions of meteorological changes to surface MDA8 O$_3$, comparing 03E15M and 03E03M (2003 standard) simulations; (b) Contributions of emission changes to surface MDA8 O$_3$, comparing 15E03M and 03E03M (2003 standard) simulations; (c) Contributions of meteorological changes to surface MDA8 O$_3$, comparing 15E15M (2015 standard) and 15E03M simulations; (d) Contributions of emission changes to surface MDA8 O$_3$, comparing 15E15M (2015 standard) and 03E15M simulations.

*Page 14 Line 25: What is difference between transboundary and long-distance transport here, and how the authors draw this conclusion based on this work?*

**Response:** we have changed "transboundary transport" and "long-distance transport" to "transport" in the revised manuscript. We found that large-scale regional transport is an important contributor to the spatial distributions and inter-annual variations of surface O$_3$

over the CEC region through the $O_3$ transport flux analysis. The original statement has been modified as follows.

"Transport issues in local $O_3$ control strategies should go beyond transport from neighbouring areas (e.g., cities) and account for the long-distance transport (e.g., across provinces)."

*Page 2 Line 8: which controls*

**Response:** changed

*Page 14 Line 2: "Asia" should be "Asian"*

**Response:** changed.